# Early anteroposterior regionalisation of human neural crest is shaped by a pro-mesodermal factor

Antigoni Gogolou[1,2], Celine Souilhol[1,2], Ilaria Granata[3], Filip J Wymeersch[4], Ichcha Manipur[3], Matthew Wind[1,2], Thomas JR Frith[1,2], Maria Guarini[5], Alessandro Bertero[6], Christoph Bock[5,7], Florian Halbritter[8], Minoru Takasato[4,9], Mario R Guarracino[10], Anestis Tsakiridis[1,2]*

[1]Centre for Stem Cell Biology, School of Biosciences, University of Sheffield, Sheffield, United Kingdom; [2]Neuroscience Institute, The University of Sheffield, Western Bank, Sheffield, United Kingdom; [3]Computational and Data Science Laboratory, High Performance Computing and Networking Institute, National Research Council of Italy, Napoli, Italy; [4]Laboratory for Human Organogenesis, RIKEN Center for Biosystems Dynamics Research, 2-2-3 Minatojima-minamimachi, Chuo-ku, Kobe, Japan; [5]CeMM Research Center for Molecular Medicine, Austrian Academy of Sciences, Vienna, Austria; [6]Molecular Biotechnology Center, Department of Molecular Biotechnology and Health Sciences, University of Torino, Torino, Italy; [7]Institute of Artificial Intelligence, Center for Medical Statistics, Informatics, and Intelligent Systems, Medical University of Vienna, Vienna, Austria; [8]St. Anna Children's Cancer Research Institute, Vienna, Austria; [9]Laboratory of Molecular Cell Biology and Development, Department of Animal Development and Physiology, Graduate School of Bio-studies, Kyoto University, Kyoto, Japan; [10]University of Cassino and Southern Lazio, Cassino, Italy

*For correspondence:
a.tsakiridis@sheffield.ac.uk

Competing interest: The authors declare that no competing interests exist.

**Abstract** The neural crest (NC) is an important multipotent embryonic cell population and its impaired specification leads to various developmental defects, often in an anteroposterior (A-P) axial level-specific manner. The mechanisms underlying the correct A-P regionalisation of human NC cells remain elusive. Recent studies have indicated that trunk NC cells, the presumed precursors of childhood tumour neuroblastoma, are derived from neuromesodermal-potent progenitors of the postcranial body. Here we employ human embryonic stem cell differentiation to define how neuromesodermal progenitor (NMP)-derived NC cells acquire a posterior axial identity. We show that TBXT, a pro-mesodermal transcription factor, mediates early posterior NC/spinal cord regionalisation together with WNT signalling effectors. This occurs by TBXT-driven chromatin remodelling via its binding in key enhancers within *HOX* gene clusters and other posterior regulator-associated loci. This initial posteriorisation event is succeeded by a second phase of trunk *HOX* gene control that marks the differentiation of NMPs toward their TBXT-negative NC/spinal cord derivatives and relies predominantly on FGF signalling. Our work reveals a previously unknown role of TBXT in influencing posterior NC fate and points to the existence of temporally discrete, cell type-dependent modes of posterior axial identity control.

## Editor's evaluation

This paper presents an interesting model of a bi-phasic regulation for Hox genes in which Wnt drives HOX regulation in neuromesodermal precursors, but it does not control HOX levels in neural crest or

spinal cord cells in human cells. The paper makes an important contribution to the literature and is of general interest.

## Introduction

The neural crest (NC) is a multipotent cell population, which arises in the dorsal neural plate/non-neural ectoderm border region during vertebrate embryogenesis and generates a variety of cell types following epithelial-to-mesenchymal transition and migration through diverse routes.NC cells emerge at all levels of the anteroposterior (A-P) axis and their A-P position determines the identity of their derivatives: cranial NC produces mesoectodermal cell types (e.g. dermis, cartilage, and bone), melanocytes, neurons, and glial cells colonising the head; vagal NC cells, arising between somites 1 and 7, contribute (together with their sacral counterparts emerging at axial levels posterior to somite 28) predominantly to the enteric nervous system and include a subpopulation (termed cardiac NC) that generates various heart structures; trunk NC generates dorsal root/sympathetic neurons, adrenal chromaffin cells, and melanocytes (reviewed in *Le Douarin et al., 2004*; *Rocha et al., 2020*; *Rothstein et al., 2018*). Defects in the specification, differentiation, or migration of NC cells (known collectively as neurocristopathies) lead to a wide spectrum of serious developmental disorders, often in an axial level-specific manner (*Pilon, 2021*; *Vega-Lopez et al., 2018*). The use of human pluripotent stem cell (hPSC) differentiation offers an attractive platform for the study of human NC biology and neurocristopathies as well as an indefinite *in vitro* source of clinically relevant NC-associated cell populations for regenerative medicine applications. However, the cellular and molecular mechanisms directing the acquisition of distinct A-P axial identities by human NC cells remain largely undefined. In turn, this obviates the design of optimised hPSC differentiation protocols aiming to produce NC derivatives as well as the dissection of the links between human neurocristopathy emergence and the axial identity of the NC cells affected.

*In vivo*, the A-P patterning of the vertebrate embryonic body and its nascent cellular components relies on the coordinated action of *Hox* gene family members (arranged as paralogous groups [PG] in four distinct chromosomal clusters: A, B, C, and D). In mammals, transcriptional activation of *Hox* genes is initiated during gastrulation, within the posterior part of the embryo around the primitive streak and proceeds in a sequential manner reflecting their 3′-to-5′ genomic order, a phenomenon described as temporal collinearity or the Hox clock (*Deschamps and Duboule, 2017*; *Dollé et al., 1989*; *Izpisúa-Belmonte et al., 1991*). The Hox clock continues to operate after gastrulation, until the end of somitogenesis, in the posterior growth zone comprising the caudal lateral epiblast-late primitive streak, and later the tail bud (*Deschamps and Duboule, 2017*; *Wymeersch et al., 2019*; *Wymeersch et al., 2021*). The colinear activation of *Hox* genes in the posterior growth region is thought to be tightly coupled to the assignment of the terminal A-P coordinates of the cell lineages that make up the postcranial axis, including the NC (*Deschamps and Duboule, 2017*; *Wymeersch et al., 2019*; *Wymeersch et al., 2021*). Eventually, the terminal expression domains of *Hox* genes along the A-P axis correspond to their ordering within their chromosomal clusters so that the earliest activated, 3′ *Hox* PG members are expressed more anteriorly compared to their later-activated 5′ counterparts. In the case of NC, anterior cranial NC is *Hox*-negative, posterior cranial NC cells express *Hox* PG(1-3) genes, while vagal NC cells are marked by the expression of *Hox* PG(3-5) members and positivity for *Hox* PG(5-9) genes denotes a trunk NC character (*Rocha et al., 2020*).

The post-gastrulation posterior growth zone is marked by high levels of WNT/FGF signalling activity and harbours a pool of multipotent axial progenitors driving embryonic axis elongation in a head-tail direction. These include neuromesodermal progenitors (NMPs) that generate a large fraction of the spinal cord as well as presomitic/paraxial mesoderm, the building blocks of the musculoskeleton (reviewed in *Henrique et al., 2015*; *Wymeersch et al., 2021*). NMPs are marked by the co-expression of neural and mesodermal genes, such as those encoding the transcription factors Sox2 and Brachyury (T) (*Guillot et al., 2021*; *Henrique et al., 2015*; *Martin and Kimelman, 2012*; *Olivera-Martinez et al., 2012*; *Tsakiridis et al., 2014*), as well as a number of other posteriorly expressed genes such as *Nkx1-2, Cdx2, Tbx6*, and Hox family members (*Amin et al., 2016*; *Dias et al., 2020*; *Gouti et al., 2017*; *Guillot et al., 2021*; *Javali et al., 2017*; *Koch et al., 2017*; *Rodrigo Albors et al., 2018*; *Wymeersch et al., 2019*; *Young et al., 2009*).

Fate mapping, lineage tracing, and single-cell transcriptomics studies in vertebrate embryos have revealed that trunk NC cells, which give rise to neuroblastoma (the most common extracranial solid tumour in infants) following oncogenic transformation, are derived from NMPs (*Javali et al., 2017*; *Lukoseviciute et al., 2021*; *Shaker et al., 2021*; *Tzouanacou et al., 2009*; *Wymeersch et al., 2016*). Moreover, recent work has demonstrated that the *in vitro* generation of trunk NC cells and their derivatives from hPSCs relies on the induction of an NMP-like intermediate following stimulation of WNT/FGF signalling pathways (*Abu-Bonsrah et al., 2018*; *Faustino Martins et al., 2020*; *Frith et al., 2018*; *Frith and Tsakiridis, 2019*; *Hackland et al., 2019*; *Kirino et al., 2018*). These and other studies have pointed to a model where a generic posterior axial identity is installed early within the NMP precursors of trunk NC cells under the influence of WNT/FGF activities prior to their differentiation and commitment to an NC fate (*Frith et al., 2018*; *Metzis et al., 2018*). However, it is unclear whether induction of a developmentally plastic, Brachyury-positive state is an obligatory step in the posteriorisation of prospective NC cells, or whether trunk NC cells can still acquire a posterior axial identity without passing through a mesoderm-competent progenitor stage.

Here we dissect the links between human NMP induction and trunk NC specification using hPSC differentiation as a model. We show that disruption of NMP ontogeny via knockdown of the NMP/mesoderm regulator TBXT (the human homologue of Brachyury) abolishes the acquisition of a posterior axial identity by NC and early spinal cord cells, as well as the adoption of an NC fate. This is linked to a failure of hPSC-derived NMPs to activate properly and subsequently maintain, during their differentiation, the expression of *HOX* genes and key signalling pathway components. We demonstrate that TBXT regulates these events by directly orchestrating an open chromatin landscape in regions associated with *HOX* clusters and loci encoding pivotal posterior/NC identity determinants. Our results also indicate that, as TBXT levels gradually decrease to extinction during the differentiation of NMPs toward early spinal cord/NC cells, the control of trunk *HOX* gene expression/posteriorisation becomes WNT-independent and relies predominantly on FGF signalling activity. Collectively, our data reveal a previously unappreciated role for TBXT in prospectively shaping the A-P patterning and specification of NC cells. They also indicate the existence of two distinct phases of posterior axial identity control: (i) an early NMP-based phase that involves the TBXT/WNT-driven establishment/fixing of a HOX-positive, posterior character in uncommitted progenitors, and, (ii) a later one, which is defined by the FGF-based maintenance of a posterior axial identity within the early spinal cord/NC derivatives of NMPs.

## Results
### NC-fated axial progenitors are marked by T expression

We and others have recently shown that hPSC-derived neuromesodermal-potent axial progenitors exhibiting expression of TBXT, a well-established regulator of NMP/axial progenitor maintenance (*Amin et al., 2016*; *Gouti et al., 2014*; *Guibentif et al., 2021*; *Koch et al., 2017*; *Martin and Kimelman, 2010*; *Rashbass et al., 1991*), are the optimal source of trunk NC cells *in vitro* (*Cooper et al., 2022*; *Frith et al., 2018*; *Frith and Tsakiridis, 2019*; *Gomez et al., 2019*; *Hackland et al., 2019*). In the mouse embryo, homotopic grafting-based fate mapping has indicated that the equivalent NM-potent axial progenitors giving rise to trunk NC are located in the lateral-most caudal epiblast of somitogenesis-stage embryos (marked as 'LE' in *Figure 1*; *Wymeersch et al., 2016*). We confirmed by wholemount immunofluorescence analysis that, similar to their human *in vitro* counterparts, the LE domain in mouse is also marked by expression of T (Brachyury) immediately before the time of early posterior NC emergence (embryonic day [E] 8.75–E9.0, Theiler stage [TS] 13–14) (*Figure 1A*). Low-level expression of Tfap2a, potentially marking nascent NC cells (*Mitchell et al., 1991*; *Rothstein and Simoes-Costa, 2020*), was first detected within the posterior LE domain from E9.0 (*Figure 1* **Bg-h** vs **Bc-d**). 'Committed' NC cells, located anteriorly at the same time points, were found to also express Sox9 alongside Tfap2a (*Figure 1* **Ba-b** and **Be-f**). These data demonstrate that NC-fated axial progenitors in the mouse embryo exhibit T expression in line with the previous lineage tracing studies showing that T-positive embryonic cells contribute to the posterior neural tube and trunk NC (*Anderson et al., 2013*; *Mugele et al., 2018*; *Shaker et al., 2021*).

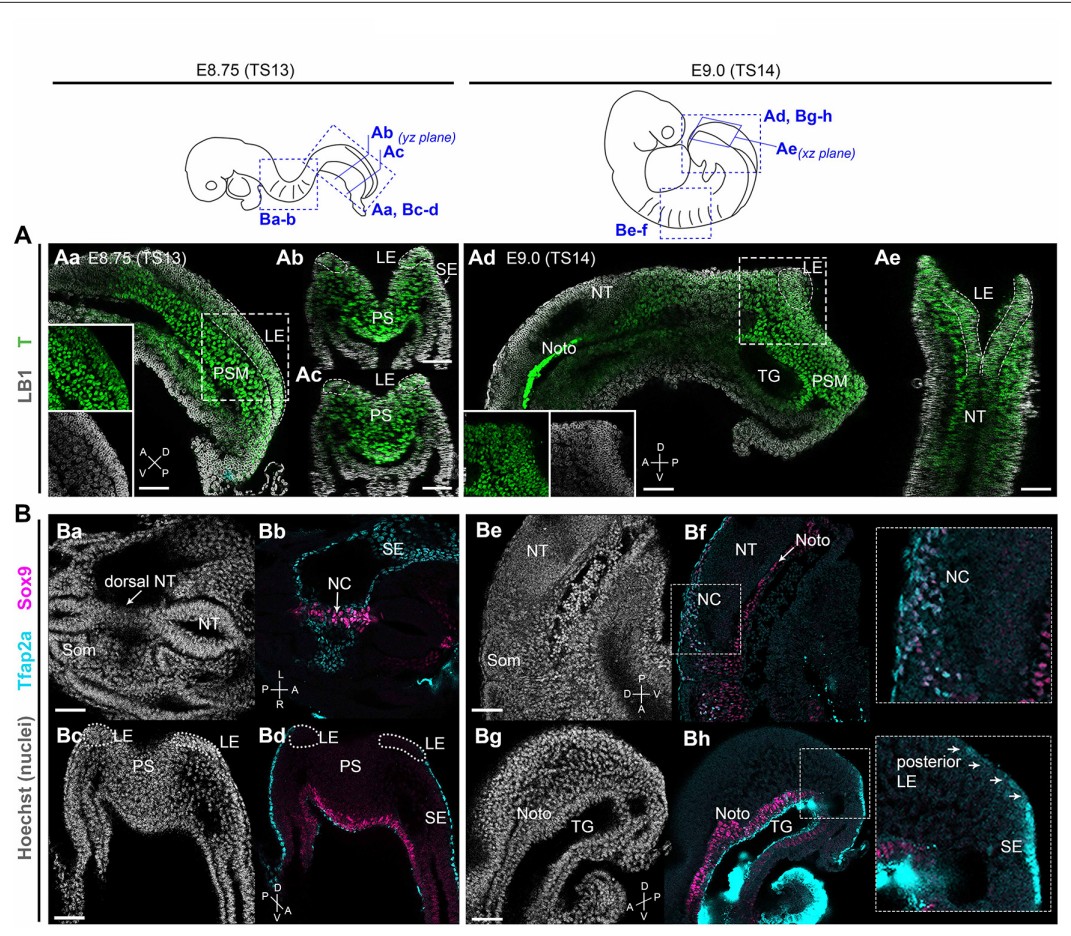

**Figure 1.** Brachyury expression marks neural crest (NC)-fated axial progenitors. Top: schematics showing location and orientation of immunostaining data in in embryonic day (E) 8.75 (Theiler stage [TS]13) and E9.0 (TS14) embryos. (**A**) Confocal sections of wholemount immunostaining showing T (Brachyury) expression (green) in the lateral-most caudal epiblast (LE, indicated by dashed lines). Anti-Lamin B1 (LB1) expression (grey) denotes nuclei. (**B**) Confocal sections of wholemount immunostaining for Sox9 (magenta) and Tfap2a (cyan) showing NC derivatives (Ba–Bb, Be–Bf) and their progenitors (Bc-Bd, Bg-Bh) in the trunk and tail bud, respectively. At E9.0, Tfap2a was detected in posterior LE progenitors and at a lower level compared to the surface ectoderm (SE; inset in Bg). Noto: notochord; NT: neural tube; PS: primitive streak; PSM: presomitic mesoderm; SE: surface ectoderm; som: somite; TG: tail gut. A: anterior; P: posterior; D: dorsal; V: ventral; L: left; R: right. Scale bars = 50 μm.

## TBXT controls posterior axial identity acquisition by NMP-derived NC cells

Our embryo analysis results suggest that the induction of trunk NC may rely on the prior generation of mesoderm-competent, T+axial progenitors emerging in the posterior growth zone during embryonic axis elongation. To test this hypothesis, we examined the effects of attenuating TBXT during human embryonic stem cell (hESC) differentiation using an hESC line engineered to exhibit short hairpin RNA (shRNA)-mediated, tetracycline (Tet)-inducible knockdown of *TBXT* expression (*Bertero et al., 2016*). We first generated NMPs from these hESCs following their treatment with the WNT agonist CHIR99021 (CHIR) and FGF2 for 3 days (*Frith et al., 2018*), in the presence or absence of Tet (*Figure 2A*). We confirmed that Tet addition induced a considerable reduction in TBXT expression both at the transcript and protein level compared to untreated controls (*Figure 2B–D*). No effect on TBXT induction was observed in NMPs generated from an isogenic control Tet-inducible shRNA hESC line targeting the *B2M* gene (*Bertero et al., 2016*) following Tet treatment (*Figure 2—figure supplement 1A*).

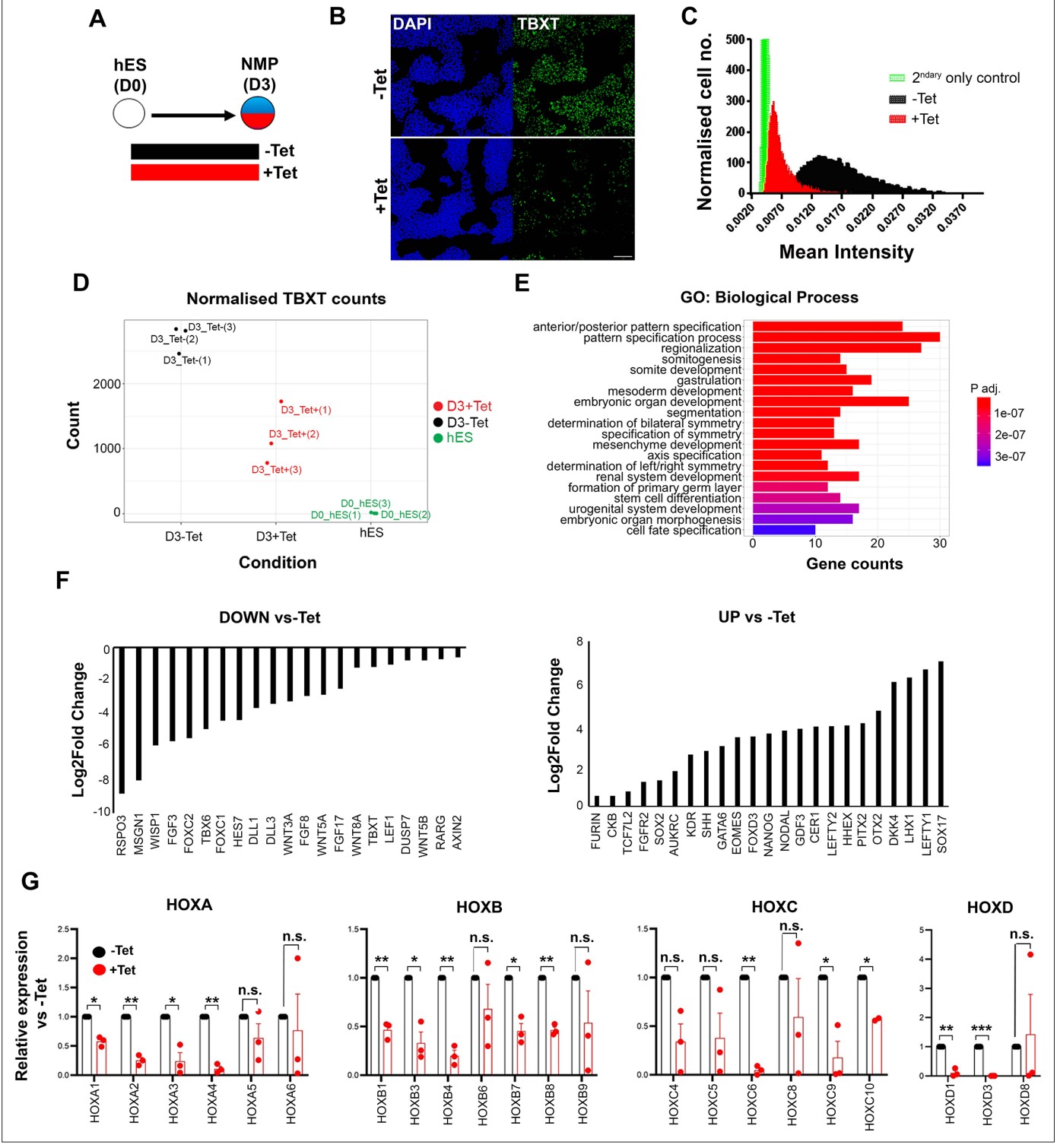

**Figure 2.** Effect of TBXT reduction on human pembryonic stem cell (hES)-derived neuromesodermal progenitors (NMPs). (**A**) Differentiation/tetracycline (Tet) treatment scheme. (**B**) Immunofluorescence analysis of the expression of TBXT in shRNA hEScell-derived NMPs in the presence and absence of Tet. Scale bar = 100 μm. (**C**) Mean fluorescence intensity of TBXT protein in Tet-treated and control NMP cultures. (**D**) Normalised expression values of *TBXT* transcripts in control, Tet-treated NMPs and undifferentiated hESC samples following RNA sequencing (RNA-seq) analysis. (**E**) Gene ontology (GO) term analysis of differentially expressed genes in hESC-derived NMPs following TBXT knockdown. (**F**) Representative significantly downregulated and upregulated transcripts following TBXT depletion. (**G**) Quantitative PCR (qPCR) expression analysis of indicated *HOX* genes in control vs Tet-treated NMPs. Error bars represent SD (n=3). *p<0.05, **p<0.005, ***p<0.0005 (paired t-test). n.s. not significant.

*Figure 2 continued on next page*

*Figure 2 continued*

The online version of this article includes the following figure supplement(s) for figure 2:

**Figure supplement 1.** Effect of Tet treatment on control B2M shRNA hESC-NMPs.

To define the impact of TBXT depletion during the transition of hPSCs toward an NMP state, we carried out transcriptome analysis of Tet-treated and control hESCs cultured in NMP-inducing conditions (i.e. presence of CHIR and FGF2) for 3 days, using RNA sequencing (RNA-seq). We found that 346 and 293 genes were significantly up- and down-regulated respectively in TBXT knockdown cells compared to untreated controls (padj<0.05, Wald test; log2FC>|0.5|; *Supplementary file 1*). Gene ontology (GO) biological processes enrichment analysis revealed that most differentially expressed genes were established regulators of A-P regionalisation/posterior development (*Figure 2E*, *Supplementary file 2*). Significantly downregulated genes included presomitic mesoderm-associated transcription factors (*TBX6*, *MSGN1*, and *FOXC1/2*) as well as WNT (e.g. *RSPO3*, *WISP1*, *WNT5A/B*, *WNT8A*, and *LEF1*), FGF (*FGF3/8/17* and *DUSP7*), and NOTCH (*HES7* and *DLL1/3*) signalling pathway-linked transcripts (*Figure 2F*, *Supplementary file 1*), most of which have been reported to be present in NMPs and their immediate mesodermal derivatives *in vivo* (*Guillot et al., 2021*; *Koch et al., 2017*; *Wymeersch et al., 2019*). Moreover, we found that TBXT knockdown triggered a significant reduction in the expression of various *HOX* genes belonging to anterior, central, and posterior PG(1-9) (*Figure 2G*, *Supplementary file 1*) while no effect was observed in Tet-treated *B2M* shRNA hESC-derived NMPs (*Figure 2—figure supplement 1B*) ruling out the possibility that the observed decrease in *HOX* transcript levels may be due to nonspecific effects of Tet and/or interference of shRNAs with the microRNA processing machinery. On the contrary, the most highly upregulated genes included anterior visceral endoderm (AVE)/endoderm, anterior neurectoderm, and pluripotency/early post-implantation epiblast-associated genes such as *SOX17*, *LEFTY1/2*, *OTX2*, *CER1*, *HHEX*, *NANOG*, and *GDF3* (*Figure 2F*, *Supplementary file 1*). Taken together, these results indicate that TBXT knockdown severely impairs the induction of NMPs and their immediate presomitic mesoderm progenitor derivatives from hPSCs. They also suggest that TBXT directs the establishment of a posteriorising signalling environment associated with early *HOX* gene activation, upon pluripotency exit, since in its absence hPSCs adopt an identity that resembles the anterior epiblast despite the combined presence of caudalising WNT (CHIR) and FGF (FGF2) signalling agonists.

We next assessed the effect of TBXT disruption/failure to induce NMPs/presomitic mesoderm on trunk NC specification. To this end, we attempted to generate trunk NC cells from inducible TBXT knockdown shRNA hESCs via an NMP intermediate. We treated cells initially with CHIR-FGF2 for 3 days to induce NMPs, followed by their replating under NC-promoting conditions (i.e. low CHIR and moderate BMP levels) for a further 5 days, as previously described (*Frith et al., 2018*; *Frith and Tsakiridis, 2019*) either in the continuous presence or absence of Tet (*Figure 3A*). We found that the expression of most *HOX* PG(1-9) genes examined was dramatically reduced in the resulting Tet-treated cultures compared to their untreated counterparts (*Figure 3B and C*) indicating that the failure of TBXT-depleted NMPs to properly induce *HOX* gene expression persists in their NC derivatives. The expression levels of *CDX2*, an NMP/spinal cord/early trunk NC regulator (*Sanchez-Ferras et al., 2016*; *Sanchez-Ferras et al., 2012*) were also found to be severely decreased (*Figure 3D*). A statistically significant reduction in the levels of some NC-associated transcripts such as *SOX10* and *PAX3* was observed in the presence of Tet (p<0.05 and p<0.01 respectively; paired t-test; *Figure 3D*) though no change in SOX10 protein levels was detected (*Figure 3E*). We also observed a trend towards a slight upregulation in the expression of the anterior NC markers *OTX2* and *ETS1* (*Simoes-Costa and Bronner, 2016*) as well as the neural progenitor marker *SOX1* (*Wood and Episkopou, 1999*) following TBXT knockdown (*Figure 3D*). However, we found no significant (paired t-test) expansion of the minor fraction of SOX1 protein-positive cells present in the trunk NC cultures, indicating that TBXT knockdown does not induce a cell identity switch from NC to CNS neurectoderm (*Figure 3—figure supplement 1*). Based on these data we conclude that the acquisition of a posterior axial identity by trunk NC cells is shaped by the action of TBXT, an NMP/pro-mesodermal regulator.

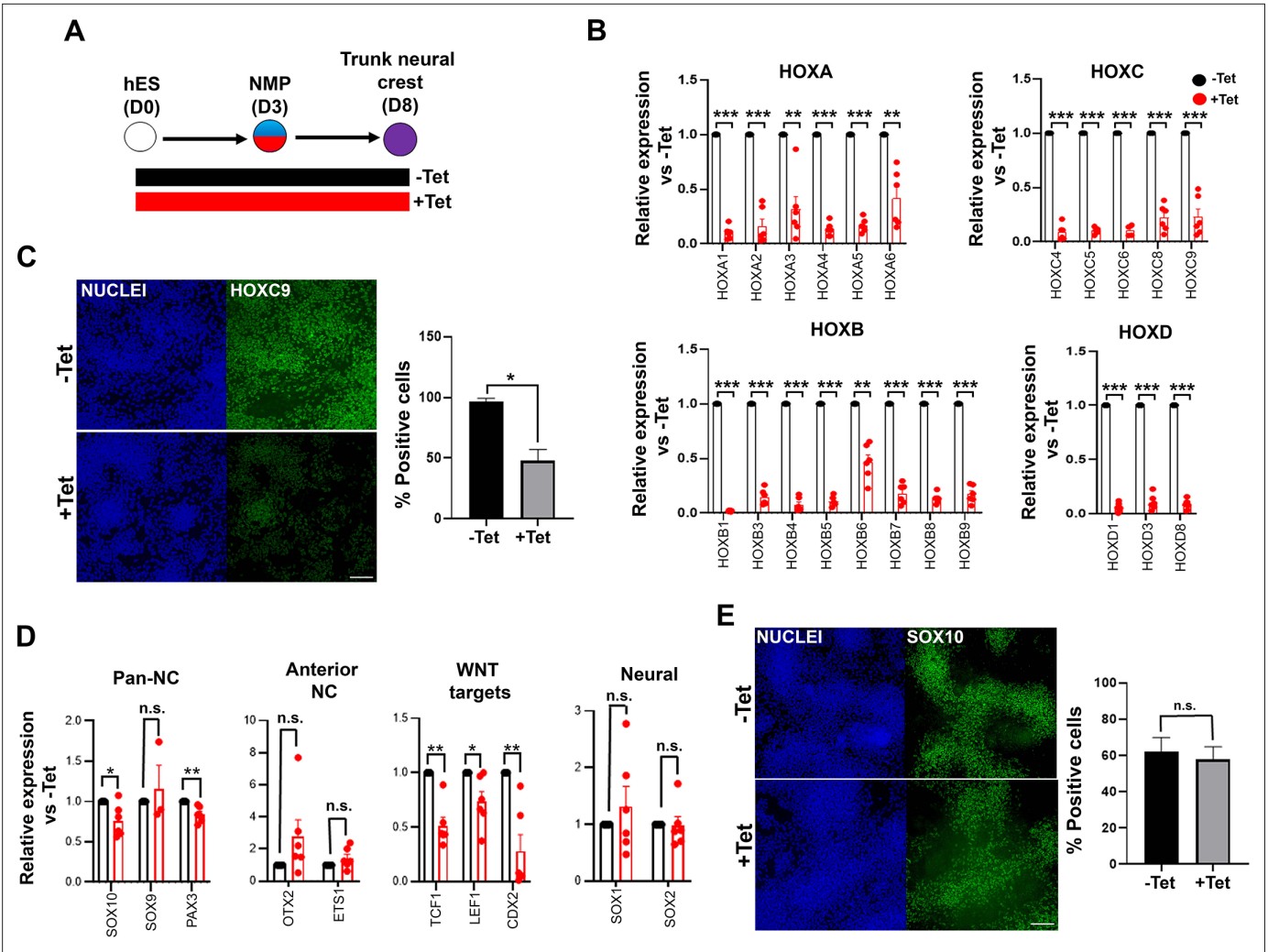

**Figure 3.** TBXT depletion impairs posterior axial identity acquisition by neural crestNC. (**A**) Differentiation/tetracycline (Tet) treatment scheme. (**B**) qPCR expression analysis of indicated *HOX* genes in control vs Tet-treated NMP-derived trunk NC cells. Error bars represent SD (n=4–6). *p<0.05, **p<0.005, ***p<0.0005 (paired t-test). (**C**) Immunofluorescence analysis of the expression of HOXC9 in control vs Tet-treated NMP-derived trunk NC cells. Quantification of HOXC9+ cells in the presence and absence of Tet is also shown. Error bars represent SD (n=3). *p<0.05 (paired t-test). Scale bar = 100 µm. (**D**) qPCR expression analysis of indicated markers in control vs Tet-treated NMP-derived trunk NC cells. Error bars represent SD (n=6). *p<0.05, **p<0.01, n.s. not significant (paired t-test). (**E**) Immunofluorescence analysis of the expression of SOX10 in control vs Tet-treated NMP-derived trunk NC cells. Quantification of SOX10+ cells in the presence and absence of Tet is also shown. Error bars represent SD (n=3). Scale bar = 100 µm. n.s. not significant (paired t-test).

The online version of this article includes the following figure supplement(s) for figure 3:

**Figure supplement 1.** TBXT depletion does not influence the number of neurectodermal cells in trunk NC cultures.

## TBXT-driven programming of a posterior NC axial identity is an early event

To further dissect the temporal role of TBXT in controlling *HOX* gene expression dynamics, we assessed the effect of manipulating its levels via Tet treatment during distinct time windows of NMP induction and differentiation (*Figure 4A*). We opted to differentiate day 3 TBXT shRNA hESC-derived NMPs in the presence of basal, serum-free media devoid of any signalling pathway agonists/antagonists for a further 4 days to minimise any potential influence of the extrinsic signalling environment on the expression status of *HOX* genes. Under these conditions, human NMPs give rise mainly to T$^{low/-}$SOX2+ posterior neural progenitors (*Gouti et al., 2014*) as well as more differentiated, mutually exclusive NC and CNS spinal cord subpopulations marked by SOX10 and SOX1 protein expression respectively (*Figure 4C and D*). We found that both early (days 0–3; NMP induction) and late (days 3–7; NMP

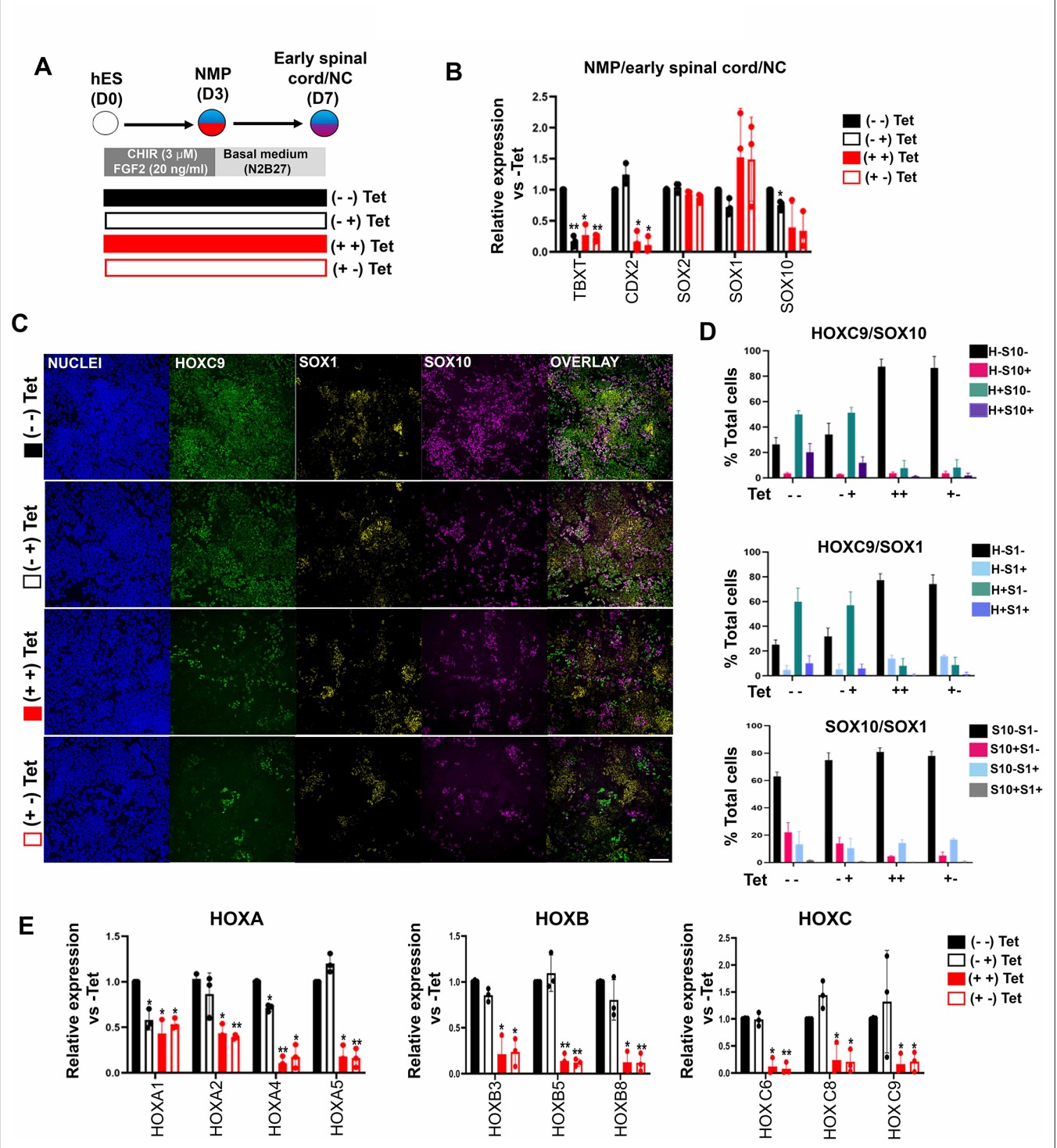

**Figure 4.** TBXT-driven programming of a posterior neural crest (NC) axial identity is an early event. (**A**) Differentiation/tetracycline (Tet) treatment scheme. The temporal Tet treatment regimens employed were: no Tet control (− −); Tet treatment between days 3 and 7 (− +); Tet treatment between days 0 and 7 (+ +); Tet treatment between days 0 and 3 (+−). (**B**) qPCR expression analysis of representative lineage identity/posterior markers in early spinal cord/NC cells generated following culture in 'neutral' basal media in the presence of Tet during the time windows shown in A. Error bars represent SD (n=3). Only statistically significant changes are indicated. *p<0.05, **p<0.005 (paired t-test). (**C**) Immunofluorescence analysis of the expression of HOXC9, SOX1, and SOX10 in early spinal cord/NC cells generated following culture in 'neutral' basal media in the presence of Tet during the time windows shown in A. Scale bar = 100 μm. Quantification of the cell populations expressing these proteins in relation to the different Tet treatment regimens is shown in (**D**). Error bars represent SD (n=3). H: HOXC9; S1: SOX1; S10: SOX10. (**E**) qPCR expression analysis of indicated *HOX*

*Figure 4 continued on next page*

*Figure 4 continued*
genes in early spinal cord/NC cells generated following culture in 'neutral' basal media in the presence of Tet during the time windows shown in A. Error bars represent SD (n=3). Only statistically significant changes are indicated. *p<0.05, **p<0.005 (paired t-test).

differentiation) time window Tet treatments resulted in a marked reduction in *TBXT* transcript levels but had a minor impact on the levels of early and definitive neural identity markers (such as *SOX2* and *SOX1*, respectively) on day 7 of differentiation (*Figure 4B*). However, we observed a statistically significant decrease in the number of cells co-expressing SOX1 and HOXC9 proteins (denoting a CNS posterior spinal cord identity) in the case of cultures treated with Tet between days 0 and 3 compared to their untreated counterparts (from ~10 to ~1%; p<0.05, paired t-test) (*Figure 4C*; compare dark blue bars in the middle graph in *Figure 4D*). A more pronounced effect of continuous (days 0–7) TBXT knockdown on both *SOX10* transcript and protein levels was also detected: the mean percentage of SOX10$^+$ cells decreased from ~23 to ~3.6% (p<0.05, paired t-test) while the mean percentage of SOX10$^+$HOXC9$^+$ cells decreased from ~20 to ~1% (p<0.05, paired t-test; *Figure 4B and C*; compare purple bars in the top graph in *Figure 4D*). The sharp decline in HOXC9 protein positivity linked to early Tet treatment (*Figure 4C and D*) was accompanied by a global downregulation in the expression of all HOX genes examined by quantitative PCR (qPCR) as well as *CDX2* (*Figure 4B and E*). On the contrary, late days 3–7 Tet treatment had a minimal effect on *HOX/CDX2* transcription (*Figure 4B and E*). Together, these data further confirm that TBXT controls the acquisition of a posterior axial identity in prospective spinal cord/NC cells and the induction of an NC fate in the absence of any extrinsic signals. Moreover, our results indicate that TBXT exerts its effect on these processes at an early stage that temporally coincides with the induction of NMPs from pluripotent cells.

## Early encoding of a posterior axial identity in NMP-derived NC cells is primarily WNT-dependent

We next sought to decipher the interplay between extrinsic signals and TBXT in orchestrating the establishment of an NC posteriorising environment. Our RNA-seq data revealed the concomitant downregulation of WNT targets and upregulation of WNT antagonists (e.g. *DKK4*) in day 3 FGF-CHIR-treated cultures in the presence of Tet (*Supplementary file 1*, *Figure 2F*) suggesting that the inability of TBXT-depleted NC cells to activate *HOX* genes may be linked to an early reduction in WNT activity. This is further supported by our observation that TBXT knockdown-triggered reduction of *HOX* gene expression in our NC cultures was also accompanied by a significant reduction in the expression of WNT signalling target genes such as *TCF1*, *LEF1*, and *CDX2* (*Figure 3D*). We, therefore, examined the temporal effects of WNT signalling perturbation on posterior axial identity acquisition during the transition from hPSCs to NMPs and subsequently NC. We first assessed the effects of WNT signalling inhibition during the 3-day induction of NMPs from hPSCs, by removing CHIR and treating the cultures with the tankyrase inhibitor XAV939 (XAV; *Huang et al., 2009*) in order to eliminate any endogenous paracrine signalling, in the presence of FGF2 (the other major signalling agonist included in the media; *Figure 5A*). Given the reported role of FGF signalling in controlling the HOX clock in NMPs and their derivatives (*Delfino-Machín et al., 2005*; *Hackland et al., 2019*; *Liu et al., 2001*; *Mouilleau et al., 2021*; *Nordström et al., 2006*), we also tested the effect of attenuating this pathway in a similar manner, by omitting FGF2 and culturing cells only in the presence of CHIR and the FGF pathway-MEK1/2 inhibitor PD0325901 (PD03) between days 0 and 3 (*Figure 5A*). Signalling inhibition was verified by confirming the downregulation/extinction of WNT (*AXIN2*, *TCF1*, and *LEF1*) and FGF signalling target genes (*SPRY4*) in inhibitor-treated D3 cultures relative to untreated controls (*Figure 5B*). Our results confirmed that maximal induction of NMP markers (*TBXT*, *CDX2*, and *NKX1-2*) and all *HOX* genes examined can be achieved only in the combined presence of WNT and FGF agonists (*Figure 5C and D*, white bars). However, WNT signalling stimulation alone, in the absence of any FGF activity, rescued the induction of most *HOX* genes and *TBXT/CDX2* with variable efficiency (*Figure 5D*, red bars) indicating that WNT is the main instructive pathway controlling global *HOX* transcription and the expression of major axis elongation regulators during the differentiation of hPSCs toward NMPs. The *HOXC* genes were an exception, as their activation was found to be equally dependent on WNT and FGF signalling (*Figure 5D*) suggesting the additional existence of *HOX* cluster-specific modes of transcriptional control, in line with similar findings in the embryo (*Ahn et al., 2014*; *van den Akker et al., 2001*).

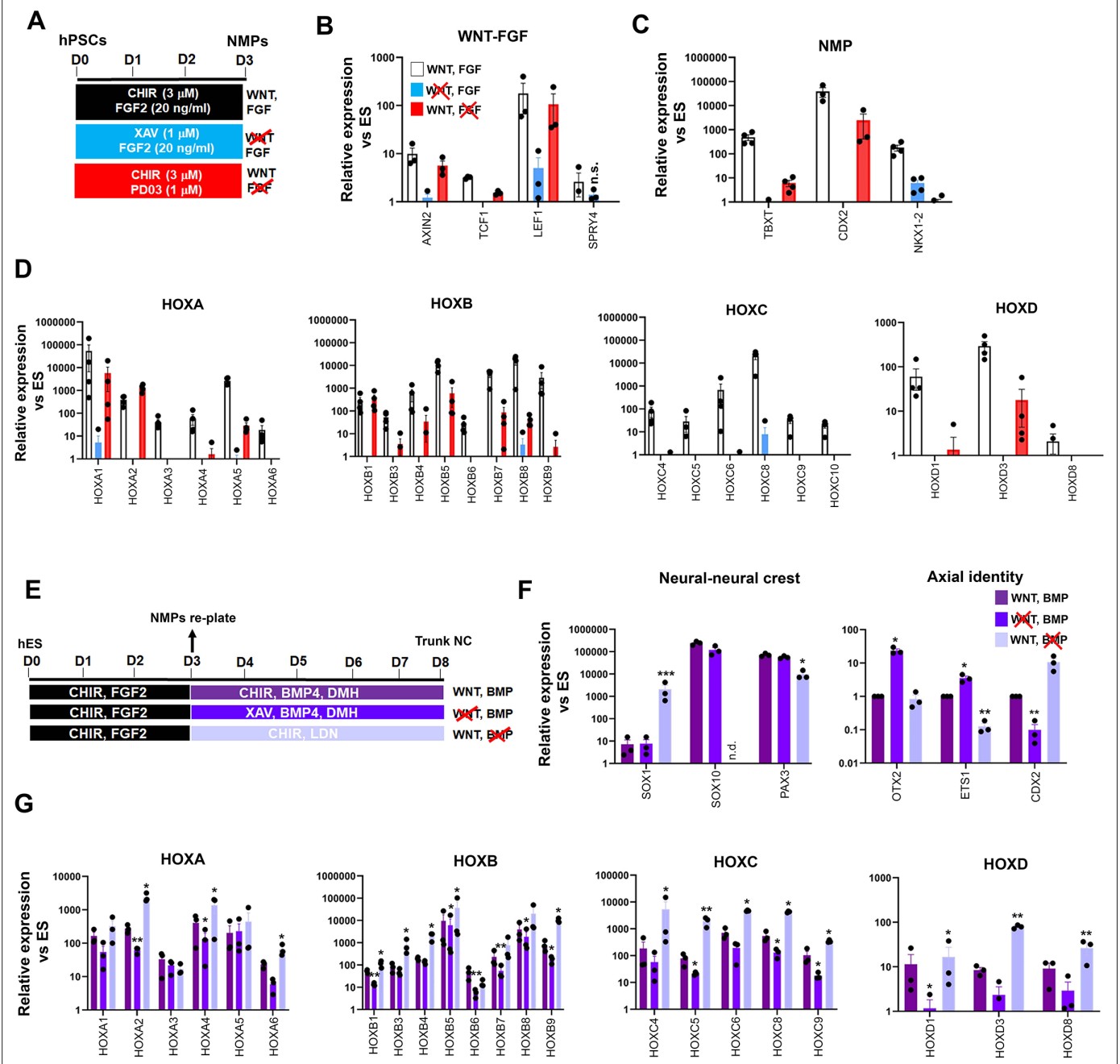

**Figure 5.** Early programming of a posterior axial identity in neuromesodermal progenitor (NMP)-derived neural crest(NC) cells is primarily WNT-dependent. (**A**) Scheme of treatments during the differentiation of human pluripotent stem cells (hPSCs) toward NMPs. (**B**) qPCR expression analysis of representative WNT-FGF targets in NMP cultures treated with the indicated combinations of WNT-FGF agonists/antagonists. Error bars represent SD (n=3–4). In all cases, changes (agonist-antagonist treatments vs corresponding WNT-FGF controls) were significantly different (ratio paired t-test) unless otherwise stated (n.s. not significant). (**C–D**) qPCR expression analysis of key NMP markers (**C**) and *HOX* genes (**D**) in NMP cultures treated with the indicated combinations of WNT-FGF agonists/antagonists. Error bars represent SD (n=3–4). In all cases, changes (agonist-antagonist treatments vs corresponding WNT-FGF controls) were significantly different (ratio paired t-test). (**E**) Scheme of treatments during the differentiation of hPSC-derived NMPs toward trunk NC cells. (**F–G**) qPCR expression analysis of representative lineage-specific, axial identity (**F**) and *HOX* genes (**G**) in NMP-derived trunk NC cultures treated with the indicated combinations of WNT-BMP agonists/antagonists. Error bars represent SD (n=3). Statistically significant changes are indicated. *p<0.05, **p<0.005.

The online version of this article includes the following figure supplement(s) for figure 5:

**Figure supplement 1.** Influence of WNT and BMP signalling on trunk NC specification.

**Figure supplement 2.** WNT signalling dynamics during posterior NC emergence.

We next interrogated the signalling pathway dependence of *HOX* gene expression during the transition of NMPs toward trunk NC cells, which involves a 5-day treatment with a lower amount of the WNT agonist CHIR and an intermediate level of BMP activity; the latter is achieved through the addition of a saturating amount of recombinant BMP4 together with the BMP type I receptor inhibitor DMH-1 to antagonistically modulate the effects of BMP4 (*Figure 5E*; *Frith et al., 2018*; *Frith and Tsakiridis, 2019*; *Hackland et al., 2017*). We examined the effect of perturbing the two main signalling pathways driving the specification of NC from NMPs (WNT and BMP) using agonist/antagonist combinations between days 3 and 8, in a manner similar to the days 0–3 treatments (*Figure 5E*). Treatment of differentiating NMPs with XAV to inhibit WNT signalling in combination with BMP stimulation resulted in a statistically significant reduction in the expression of many of the *HOX* gene family members examined albeit to a lesser degree compared to days 0–3 (*Figure 5G*, dark blue vs purple bars). No significant decrease in the overall number of cells expressing HOXC9 protein was observed (*Figure 5—figure supplement 1*). WNT signalling attenuation in the presence of BMP activity also led to a moderate increase in the expression of anterior (*OTX2* and *ETS1*) and a decrease in posterior (*CDX2*) NC markers while the levels of neural/NC-specific transcripts appeared unaffected (*Figure 5F*, dark blue vs purple bars). On the contrary, WNT stimulation in combination with the BMP inhibitor LDN193189 (LDN) resulted in significant upregulation in the expression of most *HOX* genes relative to the WNT-BMP controls (*Figure 5G*, light blue vs purple bars) with overall HOXC9 protein levels being unaffected (*Figure 5—figure supplement 1*). Moreover, this treatment resulted in the complete extinction of the expression of the definitive NC marker SOX10 and a concomitant increase in the CNS neural progenitor marker SOX1 both at the transcript and protein levels (*Figure 5F*, light blue vs purple bars; the percentage of SOX1 + cells increased from 0.7 to 45%, p<0.05 paired t-test; *Figure 5—figure supplement 1*, compare dark blue bars in the graph in *Figure 5—figure supplement 1B*) pointing to a role for BMP signalling in steering NMPs toward an NC fate at the expense of a spinal cord neurectoderm identity, in agreement with previous observations (*Leung et al., 2016*). Taken together, our results demonstrate that *HOX* gene activation in hPSC-derived NMPs and its tight coupling to the early programming of a posterior axial identity in NMP-derived NC cells are primarily driven by a WNT-TBXT regulatory loop. However, the early dependence of *HOX* gene expression on WNT signalling diminishes with time as NMPs gradually differentiate toward NC.

We went on to examine whether the reliance of early trunk NC progenitor patterning on WNT signalling occurs *in vivo*. We examined mouse *R26-WntVis* reporter embryos, in which graded nuclear EGFP expression relates to the degree of WNT signalling strength (*Takemoto et al., 2016*), at the time of posterior NC formation (E8.75–E9.0, TS13–TS14). In agreement with previous work (*Ferrer-Vaquer et al., 2010*), the highest levels of WNT activity were confined within the T+ posterior growth region (*Figure 5—figure supplement 2A*). Wholemount immunostaining followed by 3D modelling of the high anti-GFP (a-GFP)-positive fraction (a-GFP^high) demonstrated that NC-fated regions within the T+ lateral-most caudal epiblast domain are marked by higher WNT signalling levels compared to more rostral epiblast positions (dashed lines in *Figure 5—figure supplement 2B*). These findings further confirm that early specification of NM-potent T+ trunk NC progenitors correlates with high WNT activity. However, this association becomes less prevalent as cells gradually commit to a definitive NC fate in line with our *in vitro* observations showing the progressively decreasing WNT-dependence of TBXT-driven posterior axial identity programming in NC cells.

## Posterior axial identity acquisition by WNT-FGF-induced pre-neural spinal cord progenitors is TBXT-independent and FGF-dependent

Our data so far indicate that TBXT affects the induction of a posterior axial identity in NC and CNS spinal cord derivatives of NMPs in extrinsic signal-free culture conditions (*Figure 4C and D*). We have recently reported the generation of early pre-neural spinal cord progenitors following a culture of hPSC-derived NMPs under the influence of combined WNT and high FGF signalling activity; these progenitors can then be directed to efficiently generate posterior motor neurons of a thoracic axial identity (*Wind et al., 2021*). We thus examined whether TBXT knockdown disrupts *HOX* gene expression dynamics in the presence of these signals (i.e. combined WNT-high FGF). To this end, we derived early pre-neural spinal cord progenitors from TBXT shRNA hESCs: cells were treated with CHIR-FGF2 for 3 days to produce NMPs, followed by their replating and further culture in the presence of CHIR and high FGF levels for 4 days, combined with continuous (i.e. throughout the entire

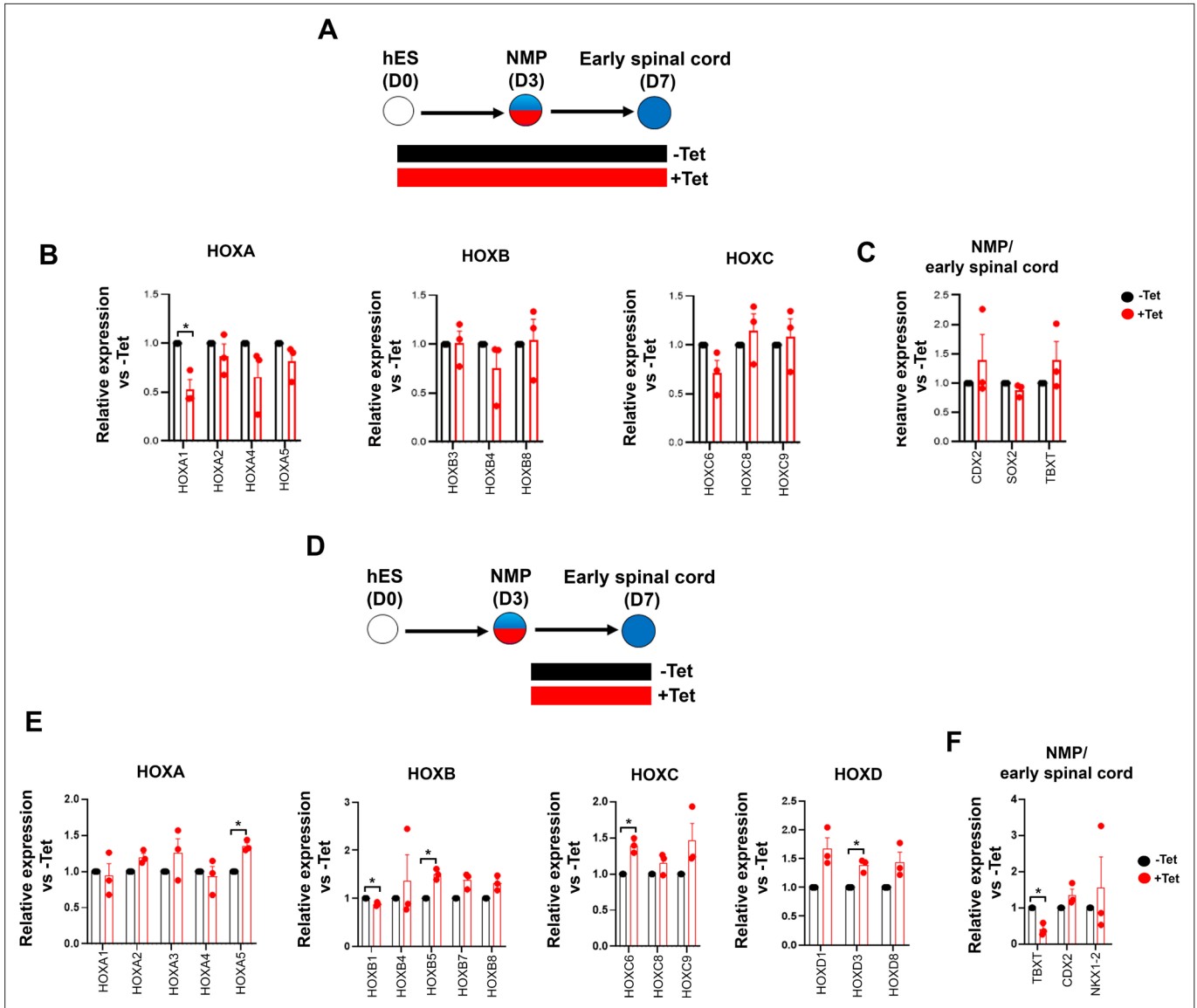

**Figure 6.** Posterior axial identity acquisition by neuromesodermal progenitor (NMP)-derived, WNT-FGF-induced pre-neural spinal cord cells is TBXT-independent. (A, D) Differentiation/treatment schemes associated with different time windows of TBXT knockdown during spinal cord differentiation from NMPs. (B–C, E–F) qPCR expression analysis of indicated *HOX* genes (B, E) and representative NMP/early spinal cord markers (C, F) in control vs tetracycline (Tet)-treated NMP-derived early spinal cord progenitors corresponding to the Tet treatment regimens shown in A and D, respectively. Error bars represent SD (n=3). Statistically significant changes are indicated. *p<0.05, n.s. not significant (paired t-test).

7-day differentiation) Tet treatment to mediate TBXT knockdown (*Figure 6A*). The levels of most *HOX* transcripts examined were largely unaffected in day 7 pre-neural spinal cord progenitors generated from TBXT-depleted cells and similar to untreated controls (*Figure 6B*). No dramatic changes in the expression of *CDX2* and *SOX2* (markers of an early pre-neural spinal cord character at this stage) were observed (*Figure 6C*). In line with our previous observations and published *in vivo* data (*Gofflot et al., 1997*; *Nordström et al., 2006*; *Wind et al., 2021*), we also detected low levels of TBXT transcripts in control cultures, and surprisingly, these were comparable to their Tet-treated counterparts (*Figure 6C*). This finding suggests that selection of cells evading shRNA knockdown and maintaining low levels of *TBXT* may occur upon culture in pre-neural spinal cord cell-promoting culture conditions. Later addition of Tet on day 3, at the start of NMP differentiation toward pre-neural spinal cord cells, appeared to restore efficient TBXT knockdown (*Figure 6D and F*) but, again, had no major impact on either *HOX* PG(1-9) gene expression or the levels of the early spinal cord/pre-neural transcripts *CDX2* and *NKX1-2* in the resulting day 7 cultures (*Figure 6E and F*). These results indicate that *HOX*

expression dynamics/posterior axial identity in NMP-derived, pre-neural spinal cord progenitors, generated under the influence of WNT/high FGF signalling activity, are largely unaffected by TBXT depletion. They also suggest that the combined presence of extrinsic WNT/FGF signals compensates for the reduced levels of TBXT, rescuing the effects of its knockdown on *HOX* gene expression in 'neutral' culture conditions (*Figure 4*).

We next investigated in more detail the role of WNT and FGF in the posteriorisation of pre-neural spinal cord progenitors since both signalling pathways have been previously implicated in the control of the *HOX* gene expression in the neural tube, through cooperation with the key axis elongation factor CDX2 (*Bel-Vialar et al., 2002*; *Lippmann et al., 2015*; *Liu et al., 2001*; *Mazzoni et al., 2013*; *Metzis et al., 2018*; *Mouilleau et al., 2021*; *Nordström et al., 2006*; *Olivera-Martinez et al., 2014*; *Takemoto et al., 2006*; *van de Ven et al., 2011*). We carried out signalling agonist/inhibitor combination experiments as described earlier (*Figure 5*) treating hPSC-derived NMPs differentiating toward pre-neural spinal cord progenitors (following their replating in the presence of CHIR-high FGF2) either with the WNT inhibitor XAV in the presence of FGF2 or the FGF/MEK inhibitor PD03 in combination with CHIR between days 3 and 7 of differentiation (*Figure 7A*). We found that WNT signalling inhibition in the presence of FGF activity during this time window does not appear to have a major effect on *HOX* or *CDX2* transcript levels (blue vs grey bars in *Figure 7B and C*). In contrast, blocking FGF signalling in combination with CHIR treatment resulted in a dramatic reduction in the expression of most *HOX* genes examined, particularly those belonging to PG(4-9), as well as *CDX2* (red bars vs grey bars, *Figure 7B and C*). The NMP/pre-neural spinal cord marker *NKX1-2* was equally affected by the two treatments while the levels of later spinal cord markers (*PAX6* and *SOX1*) were modestly elevated (*Figure 7C*). Immunofluorescence analysis further revealed that FGF signalling attenuation in the presence of CHIR results in the elimination of SOX2+HOXC9+ (their mean percentage decreased from ~45 to ~9%; n=3 p<0.05 paired t-test; *Figure 7D*, compare purple bars in the graph in *Figure 7E*) and CDX2+HOXC9+ pre-neural spinal cord populations (*Figure 7—figure supplement 1*). Based on these data, we conclude that maintenance of a posterior axial identity in NMP-derived pre-neural early spinal cord cells is driven primarily by FGF signalling, which gradually becomes functionally equivalent to TBXT in controlling trunk *HOX* gene/CDX2 expression as NMP differentiation progresses.

## TBXT controls posterior axial identity acquisition by influencing chromatin accessibility

We next examined whether TBXT controls *HOX* gene activation and potentiates the action of posteriorising extrinsic signals in CHIR-FGF-treated hPSCs via direct genomic binding or, indirectly, by affecting the expression of other key axis elongation regulators such as CDX2. To this end, we carried out chromatin immunoprecipitation followed by high-throughput sequencing (ChIP-seq) on hESC-derived NMPs and undifferentiated hESCs (control) to map TBXT targets genome-wide. We identified 24,704 TBXT-binding regions in NMPs (*Supplementary file 3*, *Figure 8—figure supplement 1A*), a large fraction of which were located within introns (~50%) and distal intergenic (~38%) regions (*Figure 8—figure supplement 1B*), reflecting similar findings on mouse T genomic binding (*Beisaw et al., 2018*; *Tosic et al., 2019*). A small number (n=2088) of non-overlapping TBXT binding sites was also detected in undifferentiated hESCs, in line with the reported low-level expression of this transcription factor in mesoderm-biased pluripotent cells (*Stavish et al., 2020*; *Tsakiridis et al., 2014*). The GO biological processes enrichment analysis of target genes associated with TXBT binding sites around the transcriptional start site (–2000 to +500 bp; 3084 peaks, 2841 genes) revealed an overrepresentation of established developmental regulators of A-P regionalisation (Benjamini–Hochberg padj <0.05; *Figure 8—figure supplement 1C*, *Supplementary file 4*). Transcription factor motif enrichment analysis revealed that the TBXT binding regions in hPSC-derived NMPs were enriched, as expected, for the Brachyury consensus binding motif as well as other T-box binding motifs such as EOMES and TBX6 (*Figure 8A*, *Supplementary file 5*), in agreement with previous findings in the mouse demonstrating that a fraction of the genomic targets of these transcription factors are also occupied by TBXT (*Koch et al., 2017*; *Tosic et al., 2019*). Other over-represented motifs included WNT signalling effectors (LEF1/TCF3) and Homeobox genes such as CDX factors (CDX2/4) and various HOX family members (*Figure 8A*, *Supplementary file 5*) reflecting possible cooperative binding between these factors and TBXT in human NMPs. Comparison of TBXT ChIP-seq targets with our list

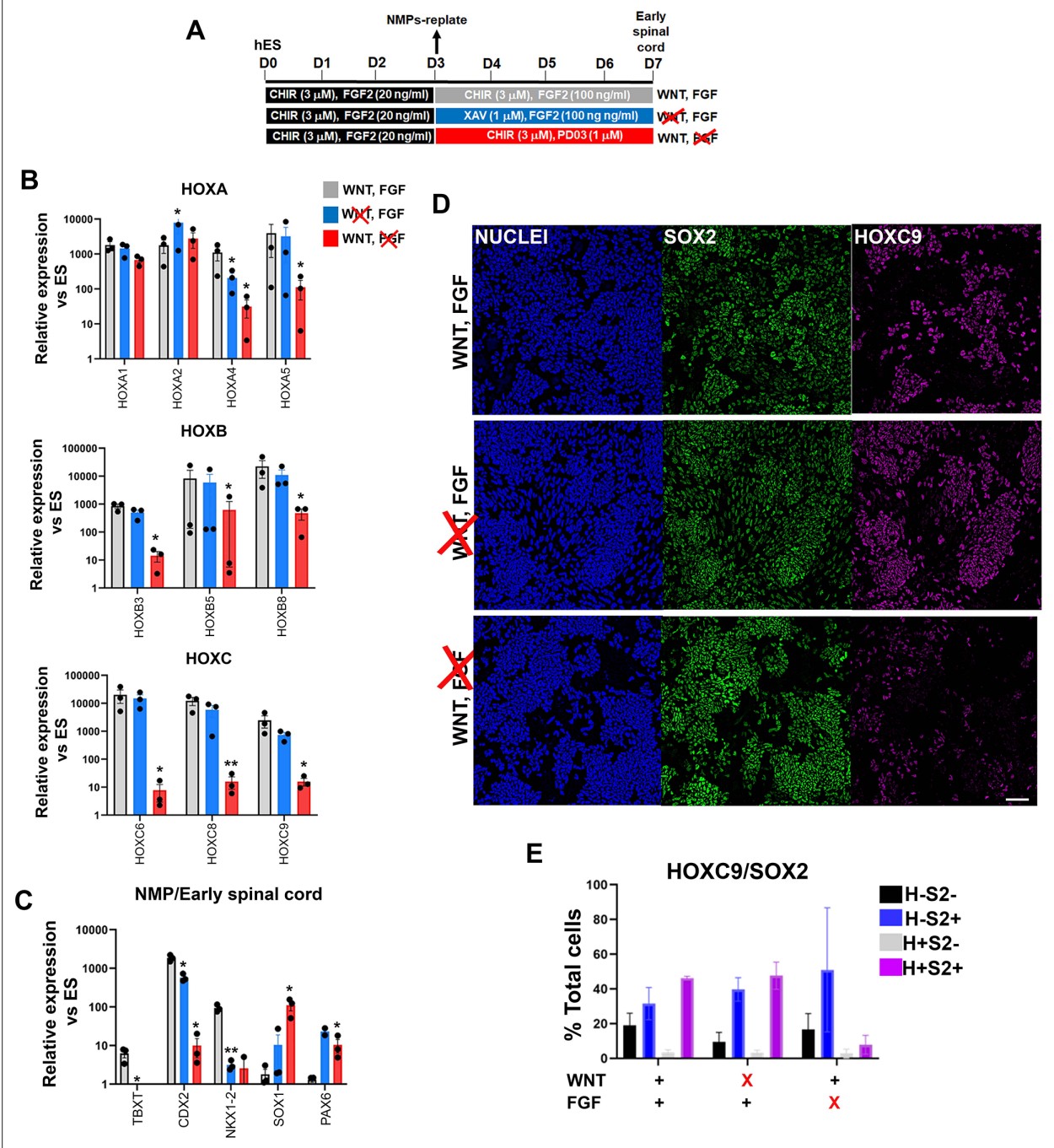

**Figure 7.** Posterior axial identity acquisition by NMP-derived pre-neural spinal cord progenitors is FGF-dependent. (**A**) Scheme of treatments during the differentiation of hPSC-derived NMPs toward early spinal cord progenitors. (**B–C**) qPCR expression analysis of indicated *HOX* genes (**B**) and representative NMP/early spinal cord/neural markers (**C**) in NMP-derived early spinal cord progenitor cultures generated as depicted in A. Error bars represent SD (n=3). Statistically significant changes are indicated. *p<0.05, **p<0.005. (**D**) Immunofluorescence analysis of SOX2 and HOXC9 protein expression in NMP-derived early spinal cord progenitor cultures treated with the indicated combinations of WNT-FGF agonists/antagonists. Quantification of the cell populations expressing these proteins in relation to the different treatment regimens is shown in (**E**). Error bars represent SD (n=3). H: HOXC9; S2: SOX2.

The online version of this article includes the following figure supplement(s) for figure 7:

**Figure supplement 1.** Posterior axial identity acquisition by NMP-derived pre-neural spinal cord progenitors is FGF-dependent.

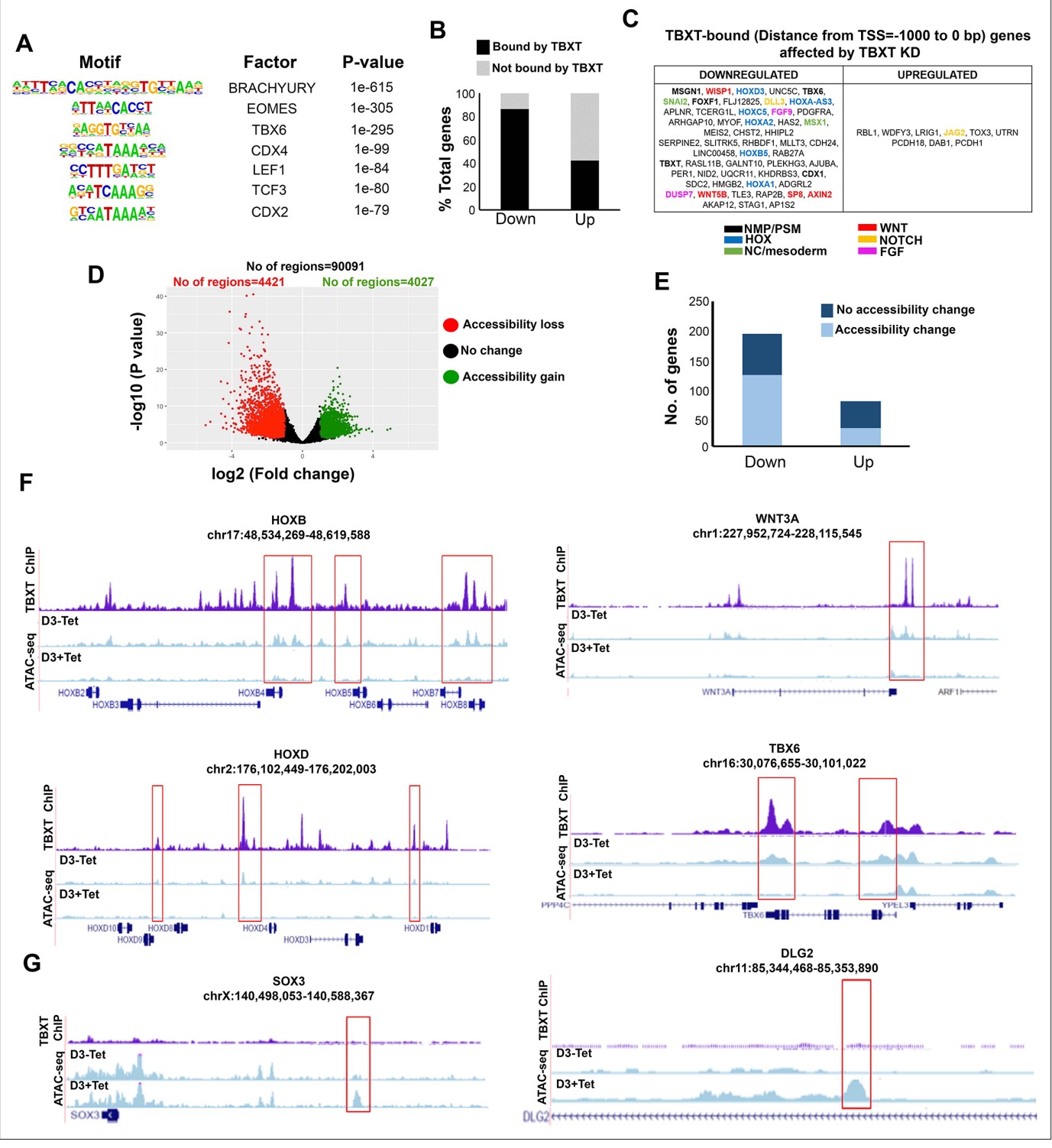

**Figure 8.** TBXT controls *HOX* gene expression and WNT signalling in NMPs by influencing chromatin accessibility. (**A**) Representative transcription factor-binding motifs enriched in TBXT binding sites. (**B**) Graph showing the percentages of differentially expressed genes (padj<0.05, log2FC>|1|) following TBXT knockdown during the transition of hESCs toward NMPs that are bound directly by TBXT. (**C**) Table showing all differentially expressed genes following TBXT knockdown that exhibit TBXT binding within their promoter region. (**D**) Volcano plot of differentially accessible ATAC-seq peaks between TBXT-depleted and control NMPs. (**E**) Graph showing the number of significantly upregulated and downregulated direct TBXT targets in relation to changes in chromatin accessibility associated with TBXT knockdown. (**F**) Correspondence between TBXT binding (ChIP) and chromatin accessibility changes (ATAC-seq) in the presence and absence of tetracycline (Tet) in indicated *HOX* clusters, WNT/presomitic mesoderm-linked loci.

*Figure 8 continued on next page*

*Figure 8 continued*

(**G**) Correspondence between TBXT binding and chromatin accessibility changes in the presence and absence of Tet in indicated neural differentiation-linked loci. Boxed areas highlight TBXT-bound regions marked by loss or gain of chromatin accessibility in the presence of Tet.

The online version of this article includes the following figure supplement(s) for figure 8:

**Figure supplement 1.** Effect of TBXT binding on chromatin accessibility.

**Figure supplement 2.** Chromatin accessibility dynamics during NMP differentiation toward trunk NC and pre-neural spinal cord progenitors.

of differentially expressed genes following Tet-induced TBXT knockdown in day 3 FGF-CHIR treated cultures (*Figure 2*) revealed that the majority (~85%) of the downregulated (padj <0.05, log2FC>|1|) genes, including HOX PG(1-9) members, pro-mesodermal factors and WNT-FGF signalling components, were directly bound by TBXT, while most (~60%) of their upregulated counterparts were not (*Figure 8B, C and F*, *Figure 8—figure supplement 1D*, *Supplementary file 6*). This indicates that TBXT acts primarily as a transcriptional activator of key downstream NMP/mesoderm regulators and *HOX* genes as previously reported in the mouse (*Amin et al., 2016*; *Beisaw et al., 2018*; *Guibentif et al., 2021*; *Koch et al., 2017*; *Lolas et al., 2014*).

To further dissect the impact of TBXT binding, we interrogated the chromatin accessibility landscape in TBXT-depleted and control hESC-derived NMPs using ATAC-seq (assay for transposase accessible chromatin with high-throughput sequencing; *Buenrostro et al., 2013*). We detected 4421 and 4027 unique regions corresponding to loss or gain of chromatin accessibility (a readout of DNA regulatory element/enhancer activity) following Tet treatment, respectively (distance from TSS ranging from –1–1 kb, log2FC cutoff = 2 and p-value<0.05; *Figure 8D*, *Supplementary file 7*). Genes associated with loss of chromatin accessibility were predominantly linked to A-P patterning, mesoderm specification, and WNT signalling (*Supplementary file 7*). Moreover, the Tet-induced downregulation of most transcripts found to be TBXT targets (*Supplementary file 3*) was also accompanied by a significant loss of 'open' chromatin regions within the TBXT binding sites (*Figure 8E and F*, *Supplementary file 7*). These included sites that were dispersed within all *HOX* clusters as well as key presomitic mesoderm regulators (e.g. *TBX6* and *MSGN1*) and WNT signalling components (e.g. *WNT3A/8* A and *LEF1*; *Figure 8F*, *Figure 8—figure supplement 1D*). On the contrary, there was no obvious correlation between upregulation of expression following TBXT knockdown and chromatin accessibility change (*Figure 8E*, *Supplementary file 7*). Gain of accessible regions in TBXT-depleted hPSC-derived NMPs appeared to occur predominantly within/around loci linked to neural development (e.g. *SOX3* and *DLG2*; *Tao et al., 2003*; *Wood and Episkopou, 1999*), most of which were not bound directly by TBXT (*Figure 8G*, *Supplementary file 7*). Moreover, examination of transcription factor binding motifs (annotated using HOMER; *Heinz et al., 2010*; Benjamini–Hochberg padj<0.05) showed that ATAC-seq sites marked by loss of chromatin accessibility following TBXT knockdown were uniquely enriched in CDX, T-box factor and HOX binding-associated DNA sequence elements whereas the regions marked by a gain of chromatin accessibility were mainly characterised by motifs indicating binding of SOX, OCT/POU, and Forkhead family transcription factors (*Figure 8—figure supplement 1E*, *Supplementary file 8*).

Further ATAC-seq analysis of trunk NC cells generated from hESC-derived NMPs in the continuous presence (from days 0–8) and absence of Tet revealed that, like in NMPs, various key loci associated with posterior patterning such as *HOX* clusters and *CDX2* were marked by a significant loss of 'open' chromatin regions in TBXT knockdown cultures (*Figure 8—figure supplement 2A,B*; *Supplementary file 9*). We also detected a major disruption of the chromatin accessibility landscape within/around WNT/BMP/FGF signalling-linked genes in Tet-treated trunk NC cells (*Figure 8—figure supplement 2C,D*; *Supplementary file 9*). Some affected components of these pathways, such as *WNT1* and *BMP7*, have been shown to play a pivotal role in trunk NC specification and delamination (*Burstyn-Cohen et al., 2004*; *Nguyen et al., 2000*). Many of the disrupted regions, particularly *HOX*-associated enhancers, were bound by TBXT and exhibited reduced chromatin accessibility already at the NMP stage (see boxed areas in *Figure 8—figure supplement 2A*). However, we also observed a loss of trunk NC-specific accessible regions, which were not direct TBXT genomic targets in NMPs (marked by red asterisks in *Figure 8—figure supplement 2*). No major changes in chromatin accessibility were detected in Tet-treated day 7 early spinal cord cultures compared to untreated controls (*Figure 8—figure supplement 2*) reflecting the minimal impact of TBXT knockdown under these conditions

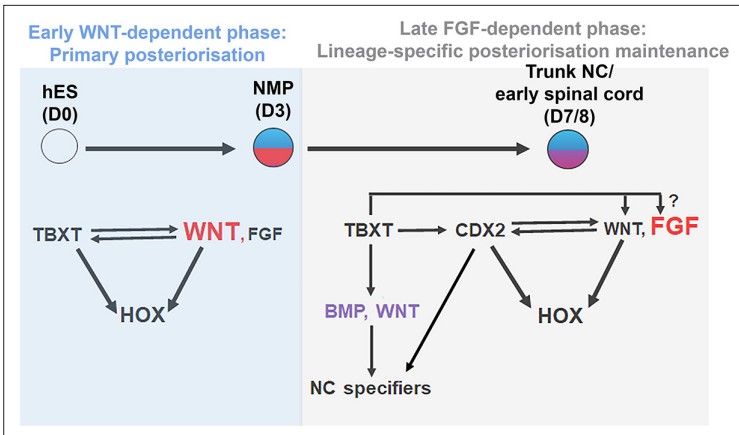

**Figure 9.** Model. Proposed model for the transcriptional and signalling control of posterior axial identity/*HOX* gene expression in hPSC derived NMPs and their derivatives.

(*Figure 6*). Collectively, these data suggest that TBXT actively reconfigures the chromatin landscape in *HOX* clusters and other axis elongation-associated loci including WNT signalling components, by direct binding to key enhancer elements to promote the acquisition of a posterior axial character/ mesoderm identity during the transition of pluripotent cells toward an NMP state. Some of these TBXT-controlled, NMP-specific 'posteriorisation' enhancers appear to remain active at later stages, in TBXT-negative trunk NC derivatives of NMPs. Moreover, our data also demonstrate the existence of trunk NC-specific enhancers that are influenced indirectly by TBXT, possibly through its binding and regulation of critical upstream transcription factors (e.g. CDX2) or signalling specifiers of NC fate during the early TBXT-positive stages of the transition of NMPs toward trunk NC.

## Discussion

A number of recent studies have demonstrated that the emergence of the trunk NC is tightly coupled to the induction of NM-potent progenitors localised in the post-gastrulation/somitogenesis stage posterior growth region of vertebrate embryos. Here we provide evidence indicating that the encoding of a posterior axial identity in human NC/early spinal cord cells, embodied by the sequential activation of *Hox* genes and the induction of posterior markers, relies on their transition through this developmentally plastic entity. This 'primary regionalisation' process (*Metzis et al., 2018*) is mediated by the action of the pro-mesodermal transcription factor TBXT, which, together with WNT signalling effectors and, possibly, CDX2, directs global *HOX* gene cluster transcription and the expression of posterior/presomitic mesoderm regulators (*Figure 9*). The early WNT dependence of *HOX* expression dynamics diminishes during the transition of NMPs towards their NC/spinal cord derivatives; this second phase of *HOX* gene expression maintenance is in turn controlled predominantly by FGF signalling (*Figure 9*). Our data also revealed another novel role of TBXT, namely its influence on the adoption of an NC but not CNS spinal cord lineage identity.

Our proposed model synthesises previous findings showing that WNT-driven posterior axial identity acquisition in neural derivatives of NMPs takes place prior to neural induction (*Lippmann et al., 2015*; *Metzis et al., 2018*; *Neijts et al., 2017*; *Neijts et al., 2016*; *Takemoto et al., 2006*) with data illustrating that temporally discrete modes of trunk axial patterning in CNS spinal cord/NC rely on both FGF and WNT activities (*Bel-Vialar et al., 2002*; *Delfino-Machín et al., 2005*; *Dunty et al., 2014*; *Hackland et al., 2019*; *Liu et al., 2001*; *Mazzoni et al., 2013*; *Mouilleau et al., 2021*; *Muhr et al., 1999*; *Nordström et al., 2006*; *Sanchez-Ferras et al., 2016*; *Sanchez-Ferras et al., 2014*; *Sanchez-Ferras et al., 2012*; *van Rooijen et al., 2012*; *Zhao et al., 2014*). Our work also reflects results showing the existence of distinct phases of *Hox* gene expression program implementation *in vivo*: (i) an early 'plastic' phase linked to A-P patterning of multipotent axial progenitors/NMPs within their posterior niche and (ii) a later phase, which marks the instalment and fixing of lineage-specific *Hox* gene expression patterns and the sharpening of their final boundaries in the neural and mesodermal derivatives of axial progenitors as and after they exit the posterior growth zone (*Ahn et al.,*

*2014*; *Brend et al., 2003*; *Charité et al., 1998*; *Deschamps and Wijgerde, 1993*; *Forlani et al., 2003*; *Hayward et al., 2015*; *McGrew et al., 2008*; *Wymeersch et al., 2019*).

We demonstrate that TBXT function is crucial for proper activation of *HOX* genes during NMP induction from hPSCs *in vitro* by channelling extrinsic WNT activity toward the establishment of a 'posteriorising'/pro-mesodermal niche (in line with previous data; *Martin and Kimelman, 2010*) that is essential for the subsequent transduction of a posterior character to NMP-derived NC/spinal cord cells. This prospective TBXT-WNT-driven transmission of positional information via an early axial progenitor intermediate and the potential inductive interaction between nascent presomitic meso- derm and NC progenitors reinforces previous observations on the effect of non-organiser mesoderm and vertical signalling as determinants of *HOX* gene expression and axial identity in early neural cells (*Bardine et al., 2014*; *Forlani et al., 2003*; *Grapin-Botton et al., 1997*). However, further work is required to disentangle the cell vs non-cell autonomous effects of TBXT on the A-P patterning of NC cells.

Our results are in line with previous studies reporting that TBXT attenuation in hPSC-derived meso- derm progenitors and spinal cord organoids, as well as in *Xenopus* morphant and mouse *Tc* (curtailed) mutant embryos abolishes *Hox* gene expression (*Faial et al., 2015*; *Libby et al., 2021*; *Lolas et al., 2014*; *Wacker et al., 2004*). Moreover, they further expand the repertoire of actions that orchestrate embryonic axis elongation/axial progenitor ontogeny and are exerted via the regulatory axis centred on WNT signalling, TBXT, and the HOX clock (*Denans et al., 2015*; *Mariani et al., 2021*; *Ye et al., 2021*; *Ye and Kimelman, 2020*). Interestingly, global *Hox* gene expression appears minimally affected in some Brachyury mutants (such as those with the $T^{2J/2J}$ genotype; *Koch et al., 2017*; *Tosic et al., 2019*), as well as in $T^{-/-}$:: wild type chimeras (*Guibentif et al., 2021*) suggesting that the nature of the *T* gene mutation and non-cell autonomous rescue effects from the surrounding wild-type environment, similar to those described previously in the case of grafted *Cdx* mutant axial progenitors (*Bialecka et al., 2010*), are critical actors in influencing the effect of Brachyury on its target genes. Moreover, the developmental arrest exhibited by *T* mutant embryos around the time of trunk NC emergence and the associated lack of posterior axial tissue precludes a detailed assessment of Hox gene expression in *T* null NC cells.

TBXT has been shown to regulate its downstream mesoderm/axis elongation-associated targets in mouse embryos by promoting chromatin remodelling and directing permissive chromatin modi- fications in key regulatory elements (*Amin et al., 2016*; *Beisaw et al., 2018*; *Koch et al., 2017*; *Tosic et al., 2019*). Our data confirm that this is also the case during the transition of hPSCs toward NMPs and reveal that TBXT additionally contributes to global control of *HOX* cluster transcription in a similar way and in line with previous reports showing direct TBXT binding in *HOX* loci during hPSC differentiation (*Faial et al., 2015*). We propose that concomitant activation of the *HOX* gene clusters and induction of WNT signalling components, via TBXT-driven chromatin landscape reconfiguration, comprise a critical early step in primary A-P regionalisation and the transition of pluripotent cells toward a caudal epiblast/axial progenitor state. This is supported by our demonstration that TBXT knockdown results in the acquisition of an anterior epiblast/AVE identity associated with WNT antag- onism/head formation-promoting activity as well as increased chromatin accessibility in regulatory elements controlling pro-neural differentiation genes. Previous data on the crucial role of Brachyury in counteracting the default neurectodermal differentiation of mouse pluripotent cells by altering chro- matin accessibility in key enhancers also point to such a model (*Tosic et al., 2019*). Moreover, early A-P regionalisation driven by the TBXT-WNT-HOX axis is likely to involve the cooperative action of CDX factors and the participation of other key transcriptional factors such as EOMES and SOX2 given their previously reported functions in the mouse (*Amin et al., 2016*; *Blassberg et al., 2022*; *Metzis et al., 2018*; *Neijts et al., 2017*; *Neijts et al., 2016*; *Tosic et al., 2019*). Thus, an additional thorough dissection of the individual roles of these factors via loss/gain of function approaches is required to elucidate their crosstalk as they fine-tune the balance between pluripotency exit, cell fate decision- making, and adoption of a posterior character in human cells.

We also provide new evidence that TBXT influences the acquisition of a SOX10-positive NC iden- tity early on during *in vitro* differentiation, a time window that coincides with the specification of NM-potent progenitors from hPSCs. This is likely to occur through two complementary routes: (i) via early direct TBXT binding and activation of DNA regulatory elements responsible for the induc- tion of key signalling and transcriptional determinants of NC fate such as WNT1, BMP7, and CDX2

(*Burstyn-Cohen et al., 2004*; *Nguyen et al., 2000*; *Sanchez-Ferras et al., 2016*; *Sanchez-Ferras et al., 2012*); (ii) via late activation of additional critical trunk NC-specific enhancers within the same TBXT-controlled NC determinants, mediated via binding of upstream transcriptional regulators induced by TBXT. This notion is supported by our finding that extrinsically supplemented BMP/WNT signals largely rescue the negative effect of TBXT knockdown on the number of SOX10$^+$ NC cells observed in 'neutral' signal agonist-free culture conditions. Unlike NC, CNS spinal cord identity induction (marked by SOX1 expression) appears to be unaffected by TBXT depletion suggesting, in line with previous studies (*Basch et al., 2006*; *Leung et al., 2016*; *Sasai et al., 2014*), that posterior NC and CNS neurectoderm lineages diverge at a pre-neural stage with TBXT acting as a potential mediator of NC fate choice at this early bifurcation point. However, further work is required to dissect the precise role of TBXT on trunk NC differentiation and assess the relevance of our hPSC-based findings in an *in vivo* context.

In summary, we provide mechanistic insight into the cellular and molecular basis of posterior axial identity acquisition during hPSC differentiation. Our data demonstrate a novel role for TBXT in controlling *HOX* gene expression and early posteriorisation supporting the idea that A-P patterning of at least some axial progenitor derivatives, such as the trunk NC, occurs prior to their specification, within their multipotent precursors. We speculate that the close links between TBXT-driven posterior axial identity programming in the NC and NMP ontogeny may explain some cases of spina bifida observed in individuals carrying mutations within the TBXT locus (*Agopian et al., 2013*; *Carter et al., 2011*; *Fellous et al., 1982*; *Morrison et al., 1996*; *Shields et al., 2000*), especially in light of the potential involvement of impaired NC specification and *HOX* gene dysregulation in neural tube defects (*Anderson et al., 2016*; *Degenhardt et al., 2010*; *Poncet et al., 2020*; *Rochtus et al., 2015*; *Yu et al., 2019*).

## Materials and methods
### Cell culture and differentiation

The use of hPSCs has been approved by the Human Embryonic Stem Cell UK Steering Committee (SCSC15-23). The *TBXT* and *B2M* shRNA sOPTiKD hESC lines (H9 background; *Bertero et al., 2016*) were employed for all TBXT loss-of-function and sequencing experiments whereas signalling agonist/antagonist treatments were performed in H9 hESCs and SFCi55-ZsGr human-induced PSCs (*Lopez-Yrigoyen et al., 2018*; *Thomson et al., 1998*). All cell lines were verified by confirming the expression of pluripotency-specific genes such as OCT4, NANOG, and TRA-1–60 using a combination of qPCR, immunofluorescence, and FACS together with human-specific primers/antibodies. Cultures were routinely monitored for bacterial and fungal contamination as well as the presence of mycoplasma. None of the cell lines is in the list of commonly misidentified cell lines maintained by the International Cell Line Authentication Committee. All cell lines were cultured routinely in feeder-free conditions in either Essential 8 (Thermo Fisher or made in-house) or mTeSR1 (Stem Cell Technologies) medium on laminin 521 (Biolamina) or Geltrex LDEV-Free reduced growth factor basement membrane matrix (Thermo Fisher). Cells were passaged twice a week after reaching approximately 80% confluency using PBS/EDTA as a passaging reagent. The TBTX inducible knockdown in the *TBXT* shRNA sOPTiKD hESC line was achieved using Tet hydrochloride (Merck Life Science) at 1 µg/ml as described previously (*Bertero et al., 2016*). The hESCs were cultured in the presence/absence of Tet for 2 days prior to the initiation of differentiation and the Tet treatment was continued throughout the differentiation for the periods indicated in the results section/schemes.

For NMP differentiation, hPSCs (70–80% confluent) were dissociated using Accutase solution (Merck Life Science) and plated at a density of approximately 50,000 cells/cm$^2$ on vitronectin (Thermo Fisher)-coated culture plates in N2B27 basal medium containing 50:50 Dulbecco's Modified Eagle's Medium (DMEM) F12 (Merck Life Science) / Neurobasal medium (Gibco) and 1 × N2 supplement (Gibco), 1 × B27 (Gibco), 1 × GlutaMAX (Gibco), 1 × Minimum Essential Medium Non-Essential Amino Acids (MEM NEAA) (Gibco), 2-Mercaptoethanol (50 µM, Gibco). The N2B27 medium was supplemented with CHIR (3 µM, Tocris), FGF2 (20 ng/ml, R&D Systems), and Rho-associated coil kinase (ROCK) inhibitor Y-27632 2HCl (10 µM, Adooq Biosciences) with the latter being withdrawn from the differentiation medium after the first day of NMP induction. For TBXT inducible knockdown, NMP medium was supplemented with 1 µg/ml Tet hydrochloride and replenished every other day. For

culture under 'neutral' conditions, control and Tet-treated NMPs were dissociated into single cells using Accutase and replated at a density of 50,000 cells/cm$^2$ on Geltrex-coated culture plates in N2B27 basal medium (made as described above for NMP differentiation) in the presence/absence of 1 µg/ml Tet hydrochloride. No small molecules were included apart from ROCK inhibitor Y-27632 2HCl (10 µM), which was added for the first 2 days to facilitate plating and increased cell survival. The basal media was replenished every other day and the cells were fixed/harvested on day 7 of differentiation for downstream analysis. For early spinal cord progenitor differentiation, day 3 hPSC-derived NMPs were dissociated into single-cell suspension using Accutase and replated at a density of 37,500 cells/cm$^2$ on Geltrex-coated culture plates in N2B27 containing FGF2 (100 ng/ml), CHIR (3 µM), and ROCK inhibitor Y-27632 2HCl (10 µM; for the first day only) in the presence or absence of Tet hydrochloride (1 µg/ml). The medium was replaced every other day until day 7 of differentiation. For trunk NC differentiation, day 3 hPSC-derived NMPs were dissociated using Accutase and replated at a density of 50,000 cells/cm$^2$ on Geltrex-coated culture plates directly into NC inducing medium consisting of DMEM/F12, 1 × N2 supplement, 1 × GlutaMAX, 1 × MEM NEAA, the TGF-beta/Activin/Nodal inhibitor SB-431542 (2 µM, Tocris), CHIR (1 µM), BMP4 (15 ng/ml, Thermo Fisher), the BMP type-I receptor inhibitor DMH-1 (1 µM, Tocris), and ROCK inhibitor Y-27632 2HCl (10 µM). The medium was refreshed every two days without ROCK inhibitor and was supplemented with 1 µg/ml Tet hydrochloride throughout the differentiation for tet-mediated inducible TBXT knockdown. Trunk NC cells were analysed on day 8 of differentiation. The following signalling pathway inhibitors were employed: the WNT antagonist tankyrase inhibitor XAV (1 µM, Tocris), the MEK1/2 inhibitor PD03 (1 µM, Merck), and LDN (100 nM, Tocris).

## Immunofluorescence and imaging

Cells were fixed in 4% Paraformaldehyde (PFA) for 10 min at room temperature, rinsed twice with PBS and permeabilised with 0.5% Triton X-100 in PBS containing 10% foetal calf serum (FCS) and 0.1% bovine serum albumin (BSA) for 15 min. Blocking was then performed in a blocking buffer consisting of 10% FCS and 0.1% BSA in PBS at room temperature for 1–2 hr or overnight at 4°C. Primary antibodies were diluted in the blocking buffer and cells were incubated with primary antibodies overnight at 4°C. Following three washes, cells were incubated with secondary antibodies conjugated to Alexa fluorophores (Invitrogen) diluted in blocking buffer for 1–2 hr at room temperature and in the dark. Cell nuclei were counterstained with Hoechst 33342 (Thermo Fisher, 1:1000) and fluorescent images were acquired using the InCell Analyser 2200 system (GE Healthcare). Images then were processed in Fiji (*Schindelin et al., 2012*) or Photoshop (Adobe) using identical brightness/contrast settings to allow comparison between different treatments. At all times, the positive/negative threshold was set using a sample incubated with secondary antibody only. The following primary antibodies were employed: anti-TBXT (Abcam, ab209665; 1:1000); anti-CDX2 (Abcam, ab76541; 1:200); anti-HOXC9 (Abcam, ab50839; 1:50); anti-SOX10 (Cell Signalling Technology, 89356; 1: 500); anti-SOX1 (R&D Systems, AF3369; 1:100); anti-SOX2 (Abcam, ab92494; 1:400).

For quantification of TBXT protein expression in NMP-like cells, nine random fields per experiment were scored (two biological replicates) and processed in the image analysis software CellProfiler (*Carpenter et al., 2006*). Nuclei were first identified using Hoechst staining and subsequently mapped onto the other fluorescent channels for single-cell fluorescence intensity quantification. Cells stained with secondary antibody only were used as a negative control to set the negative/positive threshold. A histogram was created using GraphPad Prism (GraphPad Software). For quantification of SOX10 and HOXC9 protein expression in WNT/BMP-induced trunk NC, five random fields per experiment were scored (three biological replicates) and processed as described above. For quantification of SOX1 protein levels in control vs Tet-treated NMP-derived trunk NC cells, 28 random fields per experiment were scored (three biological replicates) and processed in CellProfiler. For quantification of HOX9/SOX10, HOXC9/SOX1, and SOX10/SOX1 in 'neutral' basal media and in trunk NC cultures following WNT-BMP agonists/antagonist combinations, 25 and 30 fields per experiment were scored respectively (three biological replicates) and processed in CellProfiler using a custom pipeline. Cells were classified as single/double positive based on a pair of measurements analysis for the two channels of interest. For quantification of SOX2 and HOXC9 protein expression in WNT-FGF-induced early spinal cord progenitors, 5–10 random fields per experiment were scored (three biological replicates) and processed as described above. Graphs

were generated using GraphPad Prism (GraphPad Software), which was also employed for statistical analysis.

## Flow cytometry

For flow cytometric analysis of intracellular markers, a single cell suspension was prepared following Accutase treatment and cells were fixed in 4% PFA for 10 min at room temperature. Cells were permeabilised with 0.5% Triton X-100 in PBS containing 10% FCS and 0.1% BSA for 10 min and subsequently washed with a blocking buffer consisting of 10% FCS and 0.1% BSA in PBS. Primary antibodies diluted at the appropriate concentration in blocking buffer were added to cells followed by overnight incubation at 4°C on a shaking platform and washed off the following day. Secondary antibody diluted in blocking buffer was added for 1 hr in the dark and subsequently washed with blocking buffer. Cells were resuspended in 0.5 ml PBS for flow cytometric analysis. A separate negative control stained with secondary only and an unstained sample was used to set the gates. Flow cytometric analysis was performed using a BD LSR II flow cytometer and the data were processed using the software package FlowJo.

## Immunofluorescence analysis of wholemount embryos and 3D reconstruction

E8.75 and E9.0 (TS13-14) *R26-WntVis* embryos were dissected in the M2 medium (*Nowotschin et al., 2010*). Embryos with yolk sac and amnion removed were fixed for 25–30 min at 4°C in 4% PFA in PBS, followed by permeabilisation in 0.5% Triton X-100 in PBS for 15 min, then incubated in 0.5 M glycine in PBS in 0.1% Triton X-100 (PBST) for 20 min, after which they were washed three times in PBST and blocked overnight at 4°C in 10% serum (Merck) in PBS/0.3% Triton X-100. Both primary and Alexa Fluor-conjugated secondary antibodies (Thermo Fisher Scientific, and Alexa Fluor 647 donkey-anti-chicken (#703-606-155) from Jackson Immunoresearch; all at 2 µg/ml final concentration) were diluted in blocking buffer and samples were incubated for 48 hr on a rocking platform at 4°C. A minimum of four 25 min washes were performed with PBST on a rocking platform at room temperature after the primary and secondary antibody incubation. Antibodies used (supplier, final concentration): anti-GFP (Abcam; ab13970; 10 µg/ml); anti-Brachyury (R&D; AF2085; 1 µg/ml); anti-Lamin B1 (Abcam; ab16048; 1:800-1:1000); anti-Sox9 (Merck, AB5535; 2 µg/ml); anti-Tfap2a (DSHB, 3B5, 3.1 µg/ml). Embryos were counterstained with Hoechst33342 (Thermo Fischer Scientific; 5 µg/ml). Confocal microscopy was performed after dehydration through an increasing methanol/PBS series (25, 50, and 75%, 10 min each) and two 5 min washes in 100% methanol and clearing in 1:1 v/v methanol/benzyl alcohol:benzyl benzoate(BA:BB, 2:1; Sigma), and two washes in 100% BA:BB. Embryos were imaged in BA:BB under an LSM800 confocal system with Airyscan and GaAsP detectors (Zeiss). Two to three replicate embryos were imaged per staining and stage. Wholemount immunostaining data were processed using Zeiss or Imaris software (Oxford Instruments) using channel alignment, background subtraction and deconvolution tools. In Imaris, 3D volumes were created from single channels. These volumes represent positively stained areas. As many cells in the tail bud exhibit some level of EGFP expression, only the fraction of a-GFP-stained cells with a higher intensity (a-GFP$^{high}$) were displayed in the analyses for clarity reasons.

## Quantitative real time PCR

Total RNA was extracted using the total RNA purification kit (Norgen Biotek) following the manufacturer's instructions. The cDNA preparation was completed using the High-Capacity cDNA Reverse Transcription kit (Thermo Fisher). Quantitative real-time PCR was carried out using the QuantStudio 12 K Flex (Applied Biosystems) thermocycler in combination with the Roche UPL system and the TaqMan Fast Universal PCR Master Mix (Applied Biosystems).

Primer sequences are shown in *Supplementary file 10*. Graphs were generated using GraphPad Prism (GraphPad Software), which was also employed for statistical analysis.

## Mouse husbandry

All mice were maintained on a 12 hr-light/12 hr-dark cycle and housed at 18–23°C with 40–60% humidity. Homozygote *R26-WntVis* mice were obtained from the Laboratory for Animal Resources and Genetic Engineering at the RIKEN Center for Biosystems Dynamics Research, Kobe, Japan (accession

no. CDB0303K; RBRC10162). Homozygote *R26-WntVis* males were crossed with ICR females (JAX stock #009122, The Jackson Laboratory); all experiments were performed on heterozygote embryos. For timed matings, noon on the day of finding a vaginal plug was designated E0.5. All animal experiments were approved by the Institutional Animal Experiments Committee of RIKEN Kobe Branch (A2016-03-10). Mice were handled in accordance with the ethics guidelines of the institute.

## RNA-seq
### Sample preparation
Total RNA was harvested from day 3 NMPs obtained from *TBXT* shRNA sOPTiKD hESC in the presence and absence of Tet (three biological replicates) using the total RNA purification plus kit (Norgen BioTek) according to the manufacturer's instructions. Sample quality control, library preparation, and sequencing were carried out by Novogene (http://en.novogene.com). Library construction was carried out using the NEB Next Ultra RNA Library Prep Kit and sequencing was performed using the Illumina NoveSeq platform (PE150). Raw reads were processed through FastQC v0.11.2 and Trim Galore. Reads were aligned using STAR v2.4.2a (*Dobin et al., 2013*) to the human reference genome assembly GRCh38 (Ensembl Build 79) in the two-pass mode. The RSEM v1.3.0 (*Li and Dewey, 2011*) was used to extract expected gene counts, where genes expressed in <3 samples or with total counts ≤5 among all samples were excluded. We identified genes showing significant differential expression with DESeq2 (*Love et al., 2014*), with log2FoldChange>|0.5| and Benjamini–Hochberg-adjusted p<0.05. Data were deposited to GEO (Accession number: GSE184620).

## ChIP-seq
Chromatin immunoprecipitation followed by next-generation sequencing (ChIP-seq) was performed on approximately 10 million cells/sample (three pooled replicates per sample), fixed with 1% formaldehyde solution (11% formaldehyde, 0.1 M NaCl, 1 mM EDTA [pH 8.0], 50 mM HEPES [pH 7.9]) for 15 min at room temperature on a shaking apparatus. Fixation was quenched with 125 mM of glycine (1/20 volume of 2.5 M stock) for 5 min and then adherent cells were scraped thoroughly from the culture surface. Cells were washed, centrifuged at 800 × *g* for 10 min at 4°C and pellets were resuspended in 10 ml chilled PBS-Igepal (1 × PBS [pH 7.4], 0.5% Igepal CA-630). This pellet wash was repeated and cells were resuspended in 10 ml chilled PBS-Igepal and 1 mM phenylmethylsulfonyl fluoride (PMSF). Samples were centrifuged at 800 × *g* for 10 min at 4°C for a third time, after which the supernatant was removed and pellets were snap-frozen on dry ice and stored at –80°C. Samples were sent to Active Motif (Carlsbad, CA) for ChIP-seq. Active Motif (https://www.activemotif.com) prepared the chromatin, performed ChIP reactions, generated libraries, sequenced them, and performed basic data analysis. Chromatin was isolated by adding lysis buffer, followed by disruption with a Dounce homogenizer. Lysates were sonicated and the DNA sheared to an average length of 300–500 bp with Active Motif's EpiShear probe sonicator (cat# 53051). Genomic DNA (Input) was prepared by treating aliquots of chromatin with RNase, proteinase K, and heat for de-crosslinking, followed by SPRI beads clean up (Beckman Coulter) and quantitation by Clariostar (BMG Labtech). Extrapolation to the original chromatin volume allowed the determination of the total chromatin yield. An aliquot of chromatin (50 µg) was precleared with protein G agarose beads (Invitrogen) and genomic DNA regions of interest were isolated using 4 µg of antibody against Brachyury (R&D Systems, cat# AF2085, lot# KQP0719121). Complexes were washed, eluted from the beads with sodium dodecyl sulfate (SDS) buffer, and subjected to RNase and proteinase K treatment. Crosslinks were reversed by incubation overnight at 65°C, and ChIP DNA was purified by phenol-chloroform extraction and ethanol precipitation. qPCR reactions were carried out in triplicate on specific genomic regions using SYBR Green Supermix (Bio-Rad). The resulting signals were normalized for primer efficiency by carrying out QPCR for each primer pair using Input DNA. Illumina sequencing libraries were prepared from the ChIP and Input DNAs by the standard consecutive enzymatic steps of end-polishing, dA-addition, and adaptor ligation. Steps were performed on an automated system (Apollo 342, Wafergen Biosystems/Takara). After a final PCR amplification step, the resulting DNA libraries were quantified and sequenced on Illumina's NextSeq 500 (75 nt reads, single end). Reads were aligned to the human genome (hg38) using the BWA algorithm (default settings; *Li and Durbin, 2009*). Duplicate reads were removed and only uniquely mapped reads (mapping quality >25) were used for further analysis. Alignments were extended in silico at their 3'-ends to a length of 200 bp, which is the average genomic fragment length

in the size-selected library, and assigned to 32-nt bins along the genome. The resulting histograms (genomic 'signal maps') were stored in bigWig files. Peak locations were determined using the MACS algorithm (v2.1.0) (*Zhang et al., 2008*) with a cutoff of p-value=1e–7. Peaks that were on the ENCODE blacklist of known false ChIP-Seq peaks were removed. Signal maps and peak locations were used as input data to Active Motifs proprietary analysis program, which creates Excel detailed information on sample comparison, peak metrics, peak locations, and gene annotations. Motif analysis was carried out using the Homer software (*Heinz et al., 2010*). Regions of 200 bp surrounding the summit of the top 2500 peaks (based on MACS2 p-values) were analysed. Data were deposited to GEO (Accession number: GSE184622).

## ATAC-seq

Day 3 NMPs and day 8 trunk NC cells (50,000 cells) obtained from *TBXT* shRNA sOPTiKD hESC in the presence and absence of Tet (three biological replicates) were harvested and samples were prepared using the Illumina Tagment DNA Enzyme and Buffer Small kit (Illumina), 1% Digitonin (Promega), and EDTA-free protease inhibitor cocktail (Roche). Following DNA purification with the MinElute kit eluting in 12 µl, 1 µl of eluted DNA was used in a qPCR reaction to estimate the optimal number of amplification cycles. The remaining 11 µl of each library were amplified for the number of cycles corresponding to the Cq value (i.e. the cycle number at which fluorescence has increased above background levels) from the qPCR. Library amplification was followed by AMPure beads (Beckman Coulter) size selection to exclude fragments smaller than 150 bp and larger than 1200 bp. Library amplification was performed using custom Nextera primers. DNA concentration was measured with a Qubit fluorometer (Life Technologies) and library profile was checked with Bioanalyzer High Sensitivity assay (Agilent Technologies). Libraries were sequenced by the Biomedical Sequencing Facility at CeMM (Research Center for Molecular Medicine of the Austrian Academy of Sciences, Vienna) using the Illumina HiSeq 3000/4000 platform and the 50 bp single-end configuration. Base calling was performed by Illumina Real Time Analysis (v2.7.7) software and the base calls were converted to short reads using the IlluminaBasecallsToSam tool from the Picard toolkit (v2.19.2) ('Picard Toolkit' 2019. Broad Institute, GitHub Repository. http://broadinstitute.github.io/picard/; Broad Institute). Sequencing adapters were removed, and the low-quality reads were filtered using the fastp software (v 0.20.1) (*Chen et al., 2018*). Alignment of the short reads on GRCh38 was performed using Bowtie2 (v2.4.1) (*Langmead and Salzberg, 2012*) with the '-very-sensitive' parameter. The PCR duplicates were marked using samblaster (v0.1.24) (*Faust and Hall, 2014*), and the reads mapped to the ENCODE black-listed (*Amemiya et al., 2019*) regions were discarded prior to peak calling. To detect the open chromatin regions, we performed peak calling using the MACS2 (v2.2.7.1) (*Zhang et al., 2008*) software with the '--nomodel', '--keep-dup auto', and '--extsize 147' options. Peaks in the format of bed files were analysed for differential analysis to compare signals corresponding to the + vs –Tet samples using the GUAVA software. Differential peaks with a distance from TSS ranging from –1–1 kb, log2FC cutoff = 1 and p-value<0.05 were extracted. Finally, HOMER findMotifs (v4.11) (*Heinz et al., 2010*) was used for motif enrichment analysis over the detected open chromatin regions. Data were deposited to GEO (Accession number: GSE184227).

## Acknowledgements

We wish to thank Prof. Ludovic Vallier (University of Cambridge) for providing the *TBXT* and *B2M* shRNA sOPTiKD hESC lines and Prof. Lesley Forrester (University of Edinburgh) for providing the SFCi55-ZsGr iPSC line. We would also like to thank the Biomedical Sequencing Facility at CeMM/ Michael Schuster and Leonardo Mottta, Estelle Suaud/Active Motif for assistance with the ATAC-seq and ChIP-seq experiments respectively. Finally, we are grateful to James Briscoe, Fay Cooper, Rebecca Lea, Vicki Metzis, Matt Towers and Val Wilson for critical reading of the manuscript.

## Additional information

### Funding

| Funder | Grant reference number | Author |
| --- | --- | --- |
| Biotechnology and Biological Sciences Research Council | BB/P000444/1 | Anestis Tsakiridis |
| Horizon 2020 Framework Programme | 824070 | Anestis Tsakiridis |
| Medical Research Council | MR/V002163/1 | Anestis Tsakiridis |
| Children's Cancer and Leukaemia Group | CCLGA 2019 28 | Anestis Tsakiridis |
| Japan Society for the Promotion of Science | JP19K16157 | Filip J Wymeersch |

The funders had no role in study design, data collection and interpretation, or the decision to submit the work for publication.

### Author contributions

Antigoni Gogolou, Celine Souilhol, Formal analysis, Validation, Investigation, Visualization, Methodology, Writing – review and editing; Ilaria Granata, Data curation, Formal analysis, Validation, Investigation, Visualization, Writing – review and editing; Filip J Wymeersch, Formal analysis, Funding acquisition, Investigation, Visualization, Writing – review and editing; Ichcha Manipur, Data curation, Formal analysis, Investigation, Visualization, Writing – review and editing; Matthew Wind, Formal analysis, Investigation; Thomas JR Frith, Funding acquisition, Investigation, Writing – review and editing; Maria Guarini, Investigation; Alessandro Bertero, Resources, Writing – review and editing; Christoph Bock, Data curation, Methodology, Writing – review and editing; Florian Halbritter, Methodology, Writing – review and editing; Minoru Takasato, Mario R Guarracino, Funding acquisition, Writing – review and editing; Anestis Tsakiridis, Conceptualization, Supervision, Funding acquisition, Investigation, Visualization, Methodology, Writing – original draft, Project administration, Writing – review and editing

### Author ORCIDs

Filip J Wymeersch (ID) http://orcid.org/0000-0001-8999-4555
Christoph Bock (ID) http://orcid.org/0000-0001-6091-3088
Minoru Takasato (ID) http://orcid.org/0000-0002-0458-7414
Anestis Tsakiridis (ID) http://orcid.org/0000-0002-2184-2990

### Ethics

All animal experiments were approved by the Institutional Animal Experiments Committee of RIKEN Kobe Branch (A2016-03-10). Mice were handled in accordance with the ethics guidelines of the institute.

### Decision letter and Author response

Decision letter https://doi.org/10.7554/eLife.74263.sa1
Author response https://doi.org/10.7554/eLife.74263.sa2

## Additional files

### Supplementary files

• Supplementary file 1. Significantly up- and downregulated transcripts in Tet-treated, TBXT-depleted hESC-derived NMPs.

• Supplementary file 2. List of GO terms and corresponding gene lists enriched in Tet-treated, TBXT depleted hESC-derived NMPs.

• Supplementary file 3. List of all genomic regions (Intervals) with peak p-value below the applied threshold bound by TBXT in hESC-derived NMPs and undifferentiated hESCs.

• Supplementary file 4. List of GO terms and corresponding gene lists associated with TBXT binding sites in hPSC-derived NMPs.

• Supplementary file 5. List of known HOMER database motifs enriched in TBXT binding sites in hESC-derived NMPs.

• Supplementary file 6. List of TBXT target genes which are differentially expressed following Tet treatment (padj<0.05 log2FC>|0.5|) and Gene Ontology Biological Processes enrichment analysis. Genes are listed in relation to the genomic position (in relation to TSS) of TBXT binding within their proximity. Blue highlight denotes downregulation while red represents upregulation in expression.

• Supplementary file 7. List of ATAC-seq peaks associated with gain or loss of chromatin accessibility following TBXT depletion in hESC-derived NMPs. Gene Ontology Biological Processes enrichment analysis, list of *HOX* genes as well as other genes (padj<0.05 log2FC>|1|) affected by TBXT depletion and are associated with changes in chromatin accessibility are also included.

• Supplementary file 8. List of transcription factor DNA binding motifs enriched in ATAC-seq sites associated with chromatin accessibility gain, chromatin accessibility loss or both.

• Supplementary file 9. ATAC-seq analysis of trunk NC cells generated from hPSCs via NMPs in the presence of Tet. Includes: (i) Summary: Summary of detected peak numbers; (ii) all_peaks: List of all the peaks with *P*-value <0.05 (gained open, gained close, no change) with the relative annotations; (iii) Accessibility Loss padj <0.05: Significant Gained close peaks (pvalue adjusted (padj) <0.05); (iv) *HOX* genes: Peaks in *HOX* gene regions, yellow colour highlights the significant peaks (padj <0.05); (v) Enrichment-Accessibility Loss: Enrichment of Significant Gained close peaks (pvalue adjusted <0.05); (vi) Accessibility gain padj <0.05: Significant Gained close peaks (pvalue adjusted (padj) <0.05).

• Supplementary file 10. List of primers used.

• Transparent reporting form

## Data availability

Sequencing data have been deposited in GEO under accession codes GSE184622, GSE184620 and GSE184227.

The following datasets were generated:

| Author(s) | Year | Dataset title | Dataset URL | Database and Identifier |
|---|---|---|---|---|
| Tsakiridis A | 2021 | Early anteroposterior regionalization of human neural crest is shaped by a pro-mesodermal factor | https://www.ncbi.nlm.nih.gov/geo/query/acc.cgi?acc=GSE184227 | NCBI Gene Expression Omnibus, GSE184227 |
| Granata I, Tsakiridis A | 2021 | Early anteroposterior regionalisation of human neural crest is shaped by a pro-mesodermal factor | https://www.ncbi.nlm.nih.gov/geo/query/acc.cgi?acc=GSE184622 | NCBI Gene Expression Omnibus, GSE184622 |
| Manipur I, Granata I, Tsakiridis A | 2021 | RNA sequencing of control and TBXT-depleted human NMP-like axial progenitors | https://www.ncbi.nlm.nih.gov/geo/query/acc.cgi?acc=GSE184620 | NCBI Gene Expression Omnibus, GSE184620 |

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
