## [Editor Report]

This paper presents an interesting model of a bi-phasic regulation for Hox genes in which Wnt drives HOX regulation in neuromesodermal precursors, but it does not control HOX levels in neural crest or spinal cord cells in human cells. The paper makes an important contribution to the literature and is of general interest.

---

## [Decision Letter]

**Decision letter after peer review:**

Thank you for submitting your article "Early anteroposterior regionalisation of human neural crest is shaped by a pro-mesodermal factor" for consideration by *eLife*. Your article has been reviewed by 2 peer reviewers, and the evaluation has been overseen by a Reviewing Editor and Marianne Bronner as the Senior Editor. The reviewers have opted to remain anonymous.

Essential revisions:

The reviewing editors feel that the paper has potential but requires essential additional data to support the central claims of the paper. Moreover, the mouse work is not well integrated in the manuscript and they raise a number of questions that you would need to discuss.

The following experiments would be key:

1. Quantify the percentages of cell types that emerge in their neural crest/spinal cord differentiations with the pharmacological perturbations by immunostaining or FACS. While some markers genes are expressed throughout development and label quite different cells, the detection of specific markers in single cells will help the interpretation of their qPCR results.

2. Expand the effect of WNT inhibition on the Hox code for the spinal cord in a 7-day protocol that consists of a 3-day FGF/CHIR regime followed by basal medium treatment (as they did in PMID:25157815) to clarify the temporal and cell-type-specific effects of the HOX code via TBXT/FGF on the neural crest and/or spinal cord cells.

3. Perform TBXT ChIP-seq or ChIP-qPCR or ATAC-seq in day 7 neural crest and day 7 spinal cord cells in normal or TBXT depleted conditions to assess what is the effect of the loss of accessibility from day 3.

I refer you to the full reviews, attached below, for further details. While we welcome a resubmission, we would understand if they would rather submit elsewhere given the amount of work required.

*Reviewer #1 (Recommendations for the authors):*

The authors need to address whether the difference in NC and spinal cord cell formation maybe due to NMP heterogeneity. I am not convinced they can make the conclusion as stated.

I am not an expert on NMPs, however it strikes me that Hox genes are already expressed in the primitive streak. Is it possible that what the authors study are early epiblast / primitive streak cells rather than NMPs?

Throughout the paper the authors use a handful of marker genes to assign identity to cell populations. However, many of the genes are expressed throughout development and label quite different cells and are associated with specific cell states. It is important to acknowledge this and to use a panel of markers to define cell identity.

The PCR experiments require some further analysis and the conclusions are not always supported by the results. For example, in Figure 2 upregulation of anterior markers is not significant, but the authors conclude a 'slight upregulation'. As far as I can tell, there is no statistical analysis of several qPCR experiments.

The authors draw very strong conclusions to say that the acquisition expression of Hox genes in NMPs and the acquisition of posterior identity is primarily driven by Wnt / TBXT loop. It is possible that other pathways are involved, yet they are not tested here.

The mouse data describing Wnt activity and the expression of various markers are not well integrated into the paper. The section describing Figure 4 lacks clarity and is very difficult to follow, as is Figure 4 itself. The description in the text mixes apparently published data (without references) and results, making it difficult to disentangle what the authors wish to say. The overall conclusions are therefore difficult to follow.

*Reviewer #2 (Recommendations for the authors):*

– Immunofluorescence images lack scale bars.

– I find confusing the labels in Figure 4G and 7C as the labels state WNT when CHIR is added in the protocol, but the names of the inhibitors are left unchanged.

– It would be useful to compare the dynamics/levels of TBXT expression between the neural crest and spinal cord differentiations passed the NMP stage in the wild type and downregulation conditions.

– A recent report has identified an insertion of an Alu element on a TBXT intron that generates two TBXT isoforms (https://www.biorxiv.org/content/10.1101/2021.09.14.460388v1). Is the TBXT knock down reducing the expression of both isoforms?

– The progressive colinear and temporal activation of HOX genes as embryos extend axially is termed the HOX clock. In this manuscript, all HOX genes show a coordinated/global downregulation upon TBXT depletion experiments in the neural crest or spinal cord cells (see Figure 2B, 6B). As their analysis of HOX expression is only performed at the end point, their data does not allow for any interpretation of the temporal dynamics of the HOX clock. If they want to interpret their data as a clock, they could perform time course expression analysis of the Hox genes along the neural crest or spinal cord differentiations.

– To what extent the HOX code in human neural crest and pre-neural cells *in vitro* recapitulates the mouse in vivo dynamics? The authors could assess Hox expression in the early neural crest and pre-neural populations in their Wnt reporter mouse model.

– Is TBXT required for endogenous WNT signalling expression? Does TBXT downregulation affect Wnt signalling? The authors could look at the expression of WNT signalling genes such as AXIN2 or WNT3.

– As the authors detect a role of FGF in the induction of HOX genes in late pre-neural spinal cord progenitors, I wonder if FGF is expected to affect neural crest cells. Is there any evidence that available that shows the effect of FGF in the neural crest differentiation?

– The authors could study the changes in chromatin accessibility in early NMPs and late pre-neural cells upon WNT and FGF inhibition respectively.

– As there are spinal cord cells and neural crest cells of various posterior identities, how do the authors think the Hox code is fixed in the cells?

[Editors’ note: further revisions were suggested prior to acceptance, as described below.]

Thank you for resubmitting your work entitled "Early anteroposterior regionalisation of human neural crest is shaped by a pro-mesodermal factor" for further consideration by *eLife*. Your revised article has been evaluated by Marianne Bronner (Senior Editor).

The manuscript has been improved but there are some remaining issues raised by Reviewer 2 that I ask you to consider addressing, as outlined in the re-review below:

*Reviewer #2 (Recommendations for the authors):*

The findings of this work support an interesting model of a bi-phasic regulation for Hox genes where Wnt drives HOX regulation in NMPs, but it does not control HOX levels in the neural crest or spinal cord cells in human.

The authors have addressed the main criticisms of their work during the revision. I think the paper is ready to be published, even though I have two additional comments.

1. The mouse data does not seem to motivate accurately the question they address in the paper, and I find it distracting and unhelpful. It also makes me wonder why the authors don't do the work using mouse embryonic stem cells.

2. I would recommend the authors add the ATAC-seq of pre-neural CNS spinal cord to the manuscript. They have generated the data and it is going to be publicly accessible, so I don't see why the data is not shown. They could easily add the tracks (at least the -Tet) in their Figure 8 – supplement 2 to compare the highlighted differences in accessibility.

---

## [Author Response]

Essential revisions:The reviewing editors feel that the paper has potential but requires essential additional data to support the central claims of the paper. Moreover, the mouse work is not well integrated in the manuscript and they raise a number of questions that you would need to discuss.

To improve the integration of our in vivo data we have replaced Figure 4 and Figure 4—figure supplement 1 in the original manuscript with two simpler versions (new Figure 1 and Figure 5—figure supplement 2 in the revised manuscript) and split the related text (line numbers 271299 in original manuscript) into two thematically distinct sections (“NC-fated axial progenitors are marked by T expression” linked to figure 1 and paragraph on the high WNT activity status of TBXT^+^ trunk NC progenitors linked to Figure 5—figure supplement 1; line numbers 139-158 and 323-337 respectively in revised manuscript).

The following experiments would be key:1. Quantify the percentages of cell types that emerge in their neural crest/spinal cord differentiations with the pharmacological perturbations by immunostaining or FACS. While some markers genes are expressed throughout development and label quite different cells, the detection of specific markers in single cells will help the interpretation of their qPCR results.

We have quantified the percentages of trunk NC (identified by HOXC9 and SOX10 protein co-expression) and CNS spinal cord (indicated by HOXC9 and SOX1 protein co-expression) cells following antibody staining and image analysis to define, at the single cell level, the effect of different signalling pathway agonist/antagonist combinations on cell fate and posterior axial identity. These new data can be found in Figure 7D-E and Figure 5—figure supplement 1 in the revised manuscript.

2. Expand the effect of WNT inhibition on the Hox code for the spinal cord in a 7-day protocol that consists of a 3-day FGF/CHIR regime followed by basal medium treatment (as they did in PMID:25157815) to clarify the temporal and cell-type-specific effects of the HOX code via TBXT/FGF on the neural crest and/or spinal cord cells.

To clarify the intrinsic role of TBXT in the temporal and cell-type specific modes of *HOX* gene regulation without the interference of extrinsic signals, we carried out differentiation of day 3 Tetracycline (Tet) inducible TBXT knockdown shRNA hESC-derived NMPs in the presence of basal, serum-free media devoid of any signalling pathway agonists/antagonists for a further 4 days. These culture conditions steer NMPs towards *sox2*^+^ posterior neural progenitors and more differentiated, mutually exclusive SOX10^+^ NC and SOX1^+^ CNS spinal cord subpopulations. Temporal manipulation of TBXT expression via Tet treatment during distinct time windows (days 0-7 vs days 0-3 vs days 4-7) revealed that only early (days 0-3) TBXT knockdown results in *HOX* gene reduction and elimination of HOXC9^+^SOX10^+^ trunk NC and HOXC9^+^SOX1^+^ spinal cord progenitor populations while late TBXT knockdown has a minimal effect. We also observed an overall decrease in the levels of SOX10 protein expression under the same conditions. These new data are shown in Figure 4 and described in line numbers 226-256 in the revised manuscript.

3. Perform TBXT ChIP-seq or ChIP-qPCR or ATAC-seq in day 7 neural crest and day 7 spinal cord cells in normal or TBXT depleted conditions to assess what is the effect of the loss of accessibility from day 3.

We have added new ATAC-seq data following analysis of trunk NC cells derived from TBXT knockdown shRNA hESC-derived NMPs in the presence and absence of Tet. In line with our gene expression data (Figure 3 in revised manuscript), these show that various TBXT-bound (in NMPs) loci that are associated with posterior patterning (e.g. *HOX* genes) and WNT/BMP/FGF signalling are marked by a significant loss of ‘open’ chromatin regions in trunk NC cells following TBXT depletion. We also observed a loss of trunk NC-specific accessible regions, which are not direct TBXT genomic targets in NMPs. These data can be found in Figure 8—figure supplement 2 and Appendix Table S9 in the revised manuscript. As expected, ATAC-seq analysis of pre-neural CNS spinal cord progenitors generated from TBXT knockdown shRNA hESC-derived NMPs in the presence and absence of Tet showed no significant differences in chromatin accessibility between the two conditions again reflecting our gene expression data (Figure 6 in revised manuscript). These data were not included in the new manuscript version but they are publicly available as part of our revised GEO submission (GSE184227). Mapping of TBXT binding (e.g. by ChIP-seq) in NMPderived trunk NC cells/spinal cord progenitors is not technically possible due to the very low/absent expression of TBXT protein in these cell populations.

I refer you to the full reviews, attached below, for further details. While we welcome a resubmission, we would understand if they would rather submit elsewhere given the amount of work required.Reviewer #1 (Recommendations for the authors):The authors need to address whether the difference in NC and spinal cord cell formation maybe due to NMP heterogeneity. I am not convinced they can make the conclusion as stated.

We believe that the inclusion of new data defining the emergence of NMP derivatives at the single cell level through analysis of key trunk lineage-specific markers (HOXC9, SOX10, SOX1, *SOX2*) via immunostaining and image analysis/flow cytometry (see Figure 3—figure supplement 1, Figure 4C-D, Figure 5—figure supplement 1, Figure 7D-E in revised manuscript) should address the reviewer’s point. See also our response to the editorial comments above. It should be note that the vast majority of day 3 hESC-derived NMPs (>95%) is positive for TBXT protein expression based on antibody staining and thus the starting population for the generation of trunk NC/spinal cord progenitors can be considered largely homogeneous when it comes to the expression of this transcription factor.

I am not an expert on NMPs, however it strikes me that Hox genes are already expressed in the primitive streak. Is it possible that what the authors study are early epiblast / primitive streak cells rather than NMPs?

We presume that the reviewer refers to day 3 hESC-derived NMPs here. We have previously shown that the majority of cells in these cultures co-express TBXT and *SOX2* (Frith *et al.,* 2018; Gouti *et al.,* 2014). The simultaneous presence of these transcription factors is a reliable indicator of neuromesodermal bipotency in the post-gastrulation posterior growth zone/tail bud and not a feature of the early epiblast in either mouse or chick embryos (Guillot *et al.,* 2021; Wymeersch *et al.,* 2016). Moreover, gastrula-stage epiblast cells are marked by *Nanog* (whose expression is posteriorly confined during late gastrulation) (Osorno *et al.,* 2012) and *Otx2* (found anteriorly at later gastrulation stages) (Ang *et al.,* 1994) and none of these proteins is detected in day 3 hESC-derived NMPs (Frith *et al.*, 2018). These data, together with the demonstration that our cultures also express various other posterior markers (e.g. *CDX2*, *NKX1-2* and Hox family member belonging to paralogous groups 1-9) and can give rise to both presomitic mesoderm and spinal cord neurectoderm/NC cells under the influence of the appropriate extrinsic signal combinations (Frith *et al.*, 2018; Wind *et al.,* 2021), strongly suggest that these consist mainly of cells that resemble neuromesodermal-potent axial progenitors (which we term NMPs). However, as shown previously (Frith *et al.*, 2018; Wind *et al.*, 2021), we also expect minor subpopulations representing differentiated NMP derivatives such as presomitic mesoderm cells or non-NMP related “contaminant” cell types such as lateral plate mesoderm cells to be present in these cultures.

Throughout the paper the authors use a handful of marker genes to assign identity to cell populations. However, many of the genes are expressed throughout development and label quite different cells and are associated with specific cell states. It is important to acknowledge this and to use a panel of markers to define cell identity.

We have complemented our qPCR-based characterisation of NMP-derived cell populations with immunofluorescence/image analysis focusing on the expression of lineage-specific markers that are well characterised in the context of our differentiation experiments. Specifically, we employed SOX10 and SOX1 protein expression as readouts of a NC and spinal cord progenitor identity respectively, when tracking the emergence of these cell types during NMP differentiation by immunostaining (Figure 4C-D, Figure 5—figure supplement 1 in revised manuscript). We never detected cells co-expressing these markers at the differentiation timepoints we examined thus confirming their specificity (Figure 4C-D, Figure 5—figure supplement 1 in revised manuscript). Moreover, we employed antibody staining against CDX2 and *SOX2* to quantify pre-neural spinal cord progenitors generated from NMPs following WNT and FGF agonist treatment for 4 days (Figure 7 and Figure 7—figure supplement 1); these cultures have been shown to contain approximately 80-90% *sox2*^+^/CDX2^+^ cells and a minor (<5% of total cells) fraction of TBXT^low^-expressing cells (Wind *et al.*, 2021).

The PCR experiments require some further analysis and the conclusions are not always supported by the results. For example, in Figure 2 upregulation of anterior markers is not significant, but the authors conclude a 'slight upregulation'. As far as I can tell, there is no statistical analysis of several qPCR experiments.

We have included statistical analysis of all qPCR and immunofluorescence-based marker quantification analyses in the revised manuscript.

The authors draw very strong conclusions to say that the acquisition expression of Hox genes in NMPs and the acquisition of posterior identity is primarily driven by Wnt / TBXT loop. It is possible that other pathways are involved, yet they are not tested here.

WNT signalling is the main posteriorising pathway orchestrating the NMP-niche (Amin *et al.,* 2016; Wymeersch *et al.*, 2016; Wymeersch *et al.,* 2019) and it has been implicated in the regulation of TBXT and *Hox* gene expression (Arnold *et al.,* 2000; Martin and Kimelman, 2008; Metzis *et al.,* 2018; Yamaguchi *et al.,* 1999; Young *et al.,* 2009). Hence we decided to focus on investigating the WNT-TBXT-HOX interplay and their links with FGF/BMP signalling in the context of our *in vitro* differentiation regimens. We cannot exclude the possibility that other candidate pathways may be involved in these processes but the thorough dissection of their function is beyond the scope of this manuscript.

The mouse data describing Wnt activity and the expression of various markers are not well integrated into the paper. The section describing Figure 4 lacks clarity and is very difficult to follow, as is Figure 4 itself. The description in the text mixes apparently published data (without references) and results, making it difficult to disentangle what the authors wish to say. The overall conclusions are therefore difficult to follow.Reviewer #2 (Recommendations for the authors):– Immunofluorescence images lack scale bars.

We have included scale bars in all immunofluorescence images contained in the revised version of our manuscript.

– I find confusing the labels in Figure 4G and 7C as the labels state WNT when CHIR is added in the protocol, but the names of the inhibitors are left unchanged.

We have modified the labels in question to address the reviewer’s concern.

– It would be useful to compare the dynamics/levels of TBXT expression between the neural crest and spinal cord differentiations passed the NMP stage in the wild type and downregulation conditions.

We have previously shown that TBXT protein levels rapidly decrease during the transition of wild type hESC-derived NMPs towards pre-neural spinal cord/trunk NC cells and at the end of the differentiation cultures exhibit very low (pre-neural spinal cord) of or no (trunk NC) TBXT expression (Frith et al., 2018; Wind et al., 2021). Our qPCR data indicate that all phenotypes we observe following treatment in different culture conditions (e.g. reduction of *HOX* gene expression in trunk NC/neutral conditions or minimal effect on *HOX* transcription in WNT-FGF-based pre-neural spinal cord-promoting conditions) correspond to approximately the same extent of reduction (50-80% relative to untreated controls) of TBXT transcript levels. However, it is difficult to extract any meaningful conclusions about how TBXT dosage influences axial identity/cell fate acquisition from static snapshot-type data based on immunofluorescence/qPCR without proper lineage tracing of single cells, which is beyond the scope of this manuscript.

– A recent report has identified an insertion of an Alu element on a TBXT intron that generates two TBXT isoforms (https://www.biorxiv.org/content/10.1101/2021.09.14.460388v1). Is the TBXT knock down reducing the expression of both isoforms?

The *Alu* element described in this report is located between exons 5 and 6 and promotes alternative splicing and skipping of exon 6 to give rise to a hominid-specific *TBXT* isoform. The shRNA cassette employed in our system targets an upstream region spanning exons 3 and 4 (Bertero et al., 2016) and therefore Tet treatment should result in reduction of both wild type and *TBXT-∆Exon6* isoforms.

– The progressive colinear and temporal activation of HOX genes as embryos extend axially is termed the HOX clock. In this manuscript, all HOX genes show a coordinated/global downregulation upon TBXT depletion experiments in the neural crest or spinal cord cells (see Figure 2B, 6B). As their analysis of HOX expression is only performed at the end point, their data does not allow for any interpretation of the temporal dynamics of the HOX clock. If they want to interpret their data as a clock, they could perform time course expression analysis of the Hox genes along the neural crest or spinal cord differentiations.

We agree with the reviewer and we have removed references to the “Hox clock” in relation to our findings replacing it with “global *Hox* gene transcription” or “expression”.

– To what extent the HOX code in human neural crest and pre-neural cells *in vitro* recapitulates the mouse in vivo dynamics? The authors could assess Hox expression in the early neural crest and pre-neural populations in their Wnt reporter mouse model.

This is an interesting question but it comprises a separate project given that our focus here is on hESC differentiation. Moreover, the in vivo assessment of the expression of multiple *Hox* genes at the single cell level is a challenging task due to absence of reliable antibodies.

– Is TBXT required for endogenous WNT signalling expression? Does TBXT downregulation affect Wnt signalling? The authors could look at the expression of WNT signalling genes such as AXIN2 or WNT3.

We show that TBXT knockdown during the transition of hESCs toward NMPs and trunk NC leads to a significant decrease in the levels of WNT signalling components/targets such as *RSPO3, WISP1, WNT3A/5A/8A, LEF1, AXIN2, TCF1*. These data are included in Figures 2F, 3D and Appendix Table S1 in the revised manuscript.

– As the authors detect a role of FGF in the induction of HOX genes in late pre-neural spinal cord progenitors, I wonder if FGF is expected to affect neural crest cells. Is there any evidence that available that shows the effect of FGF in the neural crest differentiation?

See our response above regarding the potentially crucial role of FGF in the control of *HOX* gene expression in NC cells. It should be noted that in line with this concept, FGF signalling has been recently shown to influence *HOX* gene expression/axial identity acquisition in hESC-derived NC cells (Hackland *et al.*, 2019). This study is cited in our manuscript.

– The authors could study the changes in chromatin accessibility in early NMPs and late pre-neural cells upon WNT and FGF inhibition respectively.

This is a good suggestion but we opted instead to prioritise the ATAC-seq analysis of trunk NC/pre-neural spinal cord cells generated from hESCs/NMPs in the presence and absence of Tet as this would provide a greater insight into the role of TBXT, which is the focus of our study. See also our responses above.

– As there are spinal cord cells and neural crest cells of various posterior identities, how do the authors think the Hox code is fixed in the cells?

Dissection of the mechanisms orchestrating the fixing of the Hox code in spinal cord vs neural crest cells of various axial identities is a fascinating and huge topic. Based on published work, we expect that these include differential usage of cell type-specific enhancers, cross-repressive *HOX* gene interactions, synergistic co-factor binding and epigenetic/post-transcriptional/translational control of *HOX* expression to name a few.

References

Amin S, Neijts R, Simmini S, van Rooijen C, Tan SC, Kester L, van Oudenaarden A, Creyghton MP, Deschamps J (2016) Cdx and T Brachyury Co-activate Growth Signaling in the Embryonic Axial Progenitor Niche. *Cell Rep* 17: 3165-3177

Anderson MJ, Schimmang T, Lewandoski M (2016) An *FGF3*-BMP Signaling Axis Regulates Caudal Neural Tube Closure, Neural Crest Specification and Anterior-Posterior Axis Extension. *PLoS Genet* 12: e1006018

Ang SL, Conlon RA, Jin O, Rossant J (1994) Positive and negative signals from mesoderm regulate the expression of mouse Otx2 in ectoderm explants. *Development* 120: 2979-2989

Arenkiel BR, Gaufo GO, Capecchi MR (2003) Hoxb1 neural crest preferentially form glia of the PNS. *Dev Dyn* 227: 379-386

Arnold SJ, Stappert J, Bauer A, Kispert A, Herrmann BG, Kemler R (2000) Brachyury is a target gene of the Wnt/β-catenin signaling pathway. *Mech Dev* 91: 249-258

Bel S, Core N, Djabali M, Kieboom K, Van der Lugt N, Alkema MJ, Van Lohuizen M (1998) Genetic interactions and dosage effects of Polycomb group genes in mice. *Development* 125: 3543-3551 Cooper F, Gentsch GE, Mitter R, Bouissou C, Healy LE, Rodriguez AH, Smith JC, Bernardo AS (2022) Rostrocaudal patterning and neural crest differentiation of human pre-neural spinal cord progenitors *in vitro*. *Stem Cell Reports* 17: 894-910

Diaz-Cuadros M, Wagner DE, Budjan C, Hubaud A, Tarazona OA, Donelly S, Michaut A, Al Tanoury Z, Yoshioka-Kobayashi K, Niino Y *et al.* (2020) *in vitro* characterization of the human segmentation clock. *Nature* 580: 113-118

Faustino Martins JM, Fischer C, Urzi A, Vidal R, Kunz S, Ruffault PL, Kabuss L, Hube I, Gazzerro E, Birchmeier C *et al.* (2020) Self-Organizing 3D Human Trunk Neuromuscular Organoids. *Cell Stem Cell* 26: 172-186 e176

Frith TJ, Granata I, Wind M, Stout E, Thompson O, Neumann K, Stavish D, Heath PR, Ortmann D, Hackland JO *et al.* (2018) Human axial progenitors generate trunk neural crest cells *in vitro*. *eLife* 7 Glaser S, Schaft J, Lubitz S, Vintersten K, van der Hoeven F, Tufteland KR, Aasland R, Anastassiadis K, Ang SL, Stewart AF (2006) Multiple epigenetic maintenance factors implicated by the loss of Mll2 in mouse development. *Development* 133: 1423-1432

Gouti M, Delile J, Stamataki D, Wymeersch FJ, Huang Y, Kleinjung J, Wilson V, Briscoe J (2017) A Gene Regulatory Network Balances Neural and Mesoderm Specification during Vertebrate Trunk Development. *Dev Cell* 41: 243-261 e247

Gouti M, Tsakiridis A, Wymeersch FJ, Huang Y, Kleinjung J, Wilson V, Briscoe J (2014) *in vitro* generation of neuromesodermal progenitors reveals distinct roles for wnt signalling in the specification of spinal cord and paraxial mesoderm identity. *PLoS Biol* 12: e1001937

Guillot C, Djeffal Y, Michaut A, Rabe B, Pourquie O (2021) Dynamics of primitive streak regression controls the fate of neuromesodermal progenitors in the chicken embryo. *eLife* 10

Hackland JOS, Shelar PB, Sandhu N, Prasad MS, Charney RM, Gomez GA, Frith TJR, Garcia-Castro MI (2019) FGF Modulates the Axial Identity of Trunk hPSC-Derived Neural Crest but Not the Cranial-Trunk Decision. *Stem Cell Reports* 12: 920-933

Lippmann ES, Williams CE, Ruhl DA, Estevez-Silva MC, Chapman ER, Coon JJ, Ashton RS (2015) Deterministic HOX patterning in human pluripotent stem cell-derived neuroectoderm. *Stem Cell Reports* 4: 632-644

Marchal L, Luxardi G, Thome V, Kodjabachian L (2009) BMP inhibition initiates neural induction via FGF signaling and Zic genes. *Proc Natl Acad Sci U S A* 106: 17437-17442

Martin BL, Kimelman D (2008) Regulation of canonical Wnt signaling by Brachyury is essential for posterior mesoderm formation. *Dev Cell* 15: 121-133

Metzis V, Steinhauser S, Pakanavicius E, Gouti M, Stamataki D, Ivanovitch K, Watson T, Rayon T, Mousavy Gharavy SN, Lovell-Badge R *et al.* (2018) Nervous System Regionalization Entails Axial Allocation before Neural Differentiation. *Cell* 175: 1105-1118 e1117

Mouilleau V, Vaslin C, Robert R, Gribaudo S, Nicolas N, Jarrige M, Terray A, Lesueur L, Mathis MW, Croft G *et al.* (2021) Dynamic extrinsic pacing of the HOX clock in human axial progenitors controls motor neuron subtype specification. *Development* 148

Osorno R, Tsakiridis A, Wong F, Cambray N, Economou C, Wilkie R, Blin G, Scotting PJ, Chambers I, Wilson V (2012) The developmental dismantling of pluripotency is reversed by ectopic Oct4 expression. *Development* 139: 2288-2298

Stavridis MP, Lunn JS, Collins BJ, Storey KG (2007) A discrete period of FGF-induced Erk1/2 signalling is required for vertebrate neural specification. *Development* 134: 2889-2894

Wind M, Gogolou A, Manipur I, Granata I, Butler L, Andrews PW, Barbaric I, Ning K, Guarracino MR, Placzek M *et al.* (2021) Defining the signalling determinants of a posterior ventral spinal cord identity in human neuromesodermal progenitor derivatives. *Development* 148

Wymeersch FJ, Huang Y, Blin G, Cambray N, Wilkie R, Wong FC, Wilson V (2016) Position-dependent plasticity of distinct progenitor types in the primitive streak. *eLife* 5: e10042

Wymeersch FJ, Skylaki S, Huang Y, Watson JA, Economou C, Marek-Johnston C, Tomlinson SR, Wilson V (2019) Transcriptionally dynamic progenitor populations organised around a stable niche drive axial patterning. *Development* 146

Yamaguchi TP, Takada S, Yoshikawa Y, Wu N, McMahon AP (1999) T (Brachyury) is a direct target of Wnt3a during paraxial mesoderm specification. *Genes Dev* 13: 3185-3190

Ying QL, Stavridis M, Griffiths D, Li M, Smith A (2003) Conversion of embryonic stem cells into neuroectodermal precursors in adherent monoculture. *Nat Biotechnol* 21: 183-186

Young T, Rowland JE, van de Ven C, Bialecka M, Novoa A, Carapuco M, van Nes J, de Graaff W, Duluc I, Freund JN *et al.* (2009) Cdx and Hox genes differentially regulate posterior axial growth in mammalian embryos. *Dev Cell* 17: 516-526

[Editors’ note: further revisions were suggested prior to acceptance, as described below.]

Reviewer #2 (Recommendations for the authors):The findings of this work support an interesting model of a bi-phasic regulation for Hox genes where Wnt drives HOX regulation in NMPs, but it does not control HOX levels in the neural crest or spinal cord cells in human.The authors have addressed the main criticisms of their work during the revision. I think the paper is ready to be published, even though I have two additional comments.1. The mouse data does not seem to motivate accurately the question they address in the paper, and I find it distracting and unhelpful. It also makes me wonder why the authors don't do the work using mouse embryonic stem cells.

We included immunofluorescence data following analysis of E8.5-9.0 mouse embryos showing that:

i) TBXT is expressed in neural crest (NC)-fated regions in the embryonic axial progenitor niche i.e. the caudal lateral epiblast (CLE) and early tailbud (Figure 1)

ii) The same posterior NC-fated regions exhibit high levels of Wnt signalling activity (Figure 5—figure supplement 2).

These findings complement our *in vitro* observations employing hESC differentiation toward axial progenitors/trunk NC/spinal cord cells and validate their potential in vivo relevance. We are unclear about what the exact issues of reviewer #2 are with these figures/data and how mouse ESC-based experiments would be of any additional value in making the above points. I should also note that our lab is not set up for work with mESCs and it would be very difficult for us both in terms of resources/infrastructure and time investment to pursue any additional work involving this system. We estimate that we would need at least 8-10 months to obtain mESC-based data thus comprising another round of revisions.

2. I would recommend the authors add the ATAC-seq of pre-neural CNS spinal cord to the manuscript. They have generated the data and it is going to be publicly accessible, so I don't see why the data is not shown. They could easily add the tracks (at least the -Tet) in their Figure 8 – supplement 2 to compare the highlighted differences in accessibility.

We have included ATAC-seq data obtained from analysis of early spinal cord progenitors generated from TBXT knockdown and control human embryonic stem cells (hESCs) in Figure 8-Supplement 2. The raw data are also publicly available in our revised GEO submission (GSE184227).